# ADVANCING THE LOWER BOUNDS:
# AN ACCELERATED, STOCHASTIC, SECOND-ORDER METHOD WITH OPTIMAL ADAPTATION TO INEXACTNESS

**Artem Agafonov [1, 2], Dmitry Kamzolov[1], Alexander Gasnikov [2,4,5], Ali Kavis [3],**

**Kimon Antonakopoulos [3], Volkan Cevher[3], Martin Takáč[1]**

[1] Mohamed bin Zayed University of Artificial Intelligence, Abu Dhabi, UAE
[2] Moscow Institute of Physics and Technology, Dolgoprudny, Russia
[3] Laboratory for Information and Inference Systems, IEM STI EPFL, Lausanne, Switzerland
[4] Innopolis University, Kazan, Russia
[5] Skoltech, Moscow, Russia

## ABSTRACT

We present a new accelerated stochastic second-order method that is robust to both gradient and Hessian inexactness, which occurs typically in machine learning. We establish theoretical lower bounds and prove that our algorithm achieves optimal convergence in both gradient and Hessian inexactness in this key setting. We further introduce a tensor generalization for stochastic higher-order derivatives. When the oracles are non-stochastic, the proposed tensor algorithm matches the global convergence of Nesterov Accelerated Tensor method. Both algorithms allow for approximate solutions of their auxiliary subproblems with verifiable conditions on the accuracy of the solution.

## 1 INTRODUCTION

In this paper, we consider the following general convex optimization problem:
$$\min_{x \in \mathbb{R}^d} f(x), \tag{1}$$
where $f(x)$ is a convex and sufficiently smooth function. We assume that a solution $x^* \in \mathbb{R}^d$ exists and we denote $f^* := f(x^*)$. We define $R = \|x_0 - x^*\|$ as a distance to the solution.

**Assumption 1.1.** *The function $f(x) \in C^2$ has $L_2$-Lipschitz-continuous Hessian if for any $x, y \in \mathbb{R}^d$*
$$\|\nabla^2 f(x) - \nabla^2 f(y)\| \leq L_2 \|x - y\|.$$

Since the calculation of an exact gradient is very expensive or impossible in many applications in several domains, including machine learning, statistics, and signal processing, efficient methods that can work with inexact stochastic gradients are of great interest.

**Assumption 1.2.** *For all $x \in \mathbb{R}^d$, we assume that stochastic gradients $g(x, \xi) \in \mathbb{R}^d$ satisfy*
$$\mathbb{E}[g(x, \xi) \mid x] = \nabla f(x), \quad \mathbb{E}\left[\|g(x, \xi) - \nabla f(x)\|^2 \mid x\right] \leq \sigma_1^2. \tag{2}$$

Extensive research has been conducted on first-order methods, both from a theoretical and practical perspective. For $L_1$-smooth functions with stochastic gradients characterized by a variance of $\sigma_1^2$, lower bound $\Omega\left(\frac{\sigma_1 R}{\sqrt{T}} + \frac{L_1 R^2}{T^2}\right)$ has been established by Nemirovski & Yudin (1983). For $f(x) = \mathbb{E}[F(x, \xi)]$, the stochastic approximation (SA) was developed, starting from the pioneering paper by Robbins & Monro (1951). Important improvements of the SA were developed by Polyak (1990); Polyak & Juditsky (1992); Nemirovski et al. (2009), where longer stepsizes with iterate averaging and proper step-size modifications were proposed, obtaining the rate $O\left(\frac{\sigma_1 R}{\sqrt{T}} + \frac{L_1 R^2}{T}\right)$. The optimal method matching the lower bounds has been developed by Lan (2012) with a convergence rate $O\left(\frac{\sigma_1 R}{\sqrt{T}} + \frac{L_1 R^2}{T^2}\right)$. However, the literature on second-order methods is significantly limited for the study of provable, globally convergent stochastic second-order methods for convex minimization.

**Second-order methods.** Although second-order methods have been studied for centuries (Newton, 1687; Raphson, 1697; Simpson, 1740; Kantorovich, 1949; Moré, 1977; Griewank, 1981), most of the

results are connected with local quadratic convergence. The significant breakthroughs regarding global convergence have been achieved only recently, starting from the paper on Cubic Regularized Newton (CRN) method by Nesterov & Polyak (2006), the first second-order method with a global convergence rate $O\left(\frac{L_2 R^3}{T^2}\right)$. Following this work, Nesterov (2008) proposes an acceleration mechanism on top of CRN and achieves the convergence rate of $O\left(\frac{L_2 R^3}{T^3}\right)$, going beyond the $\Omega(1/T^2)$ lower bound for first-order methods. Another cornerstone in the field is the work by Monteiro & Svaiter (2013), which achieves lower complexity bound $\Omega\left(\frac{L_2 R^3}{T^{7/2}}\right)$ (Agarwal & Hazan, 2018; Arjevani et al., 2019) up to a logarithmic factor, for the first time in the literature. The gap between upper and lower bounds was closed only in 2022 in subsequent works of Kovalev & Gasnikov (2022); Carmon et al. (2022). One of the main limitations of the second-order methods is a high per-iteration cost as they require computation of exact Hessian. Therefore, it is natural to use approximations of derivatives instead of their exact values. In (Ghadimi et al., 2017), CRN method with $\delta_2$-inexact Hessian information and its accelerated version were proposed, achieving convergence rate $O\left(\frac{\delta_2 R^2}{T^2} + \frac{L_2 R^3}{T^3}\right)$. This algorithm was later extended by Agafonov et al. (2023) to handle $\delta_1$-inexact gradients (and high-order derivatives) with a resulting convergence rate of $O\left(\delta_1 R + \frac{\delta_2 R^2}{T^2} + \frac{L_2 R^3}{T^3}\right)$. A recent paper by Antonakopoulos et al. (2022) proposes a stochastic *adaptive* second-order method based on the extragradient method without line-search and with the convergence rate $O\left(\frac{\sigma_1 R}{\sqrt{T}} + \frac{\sigma_2 R^2}{T^{3/2}} + \frac{L_2 R^3}{T^3}\right)$ when gradients and Hessians are noisy with variances $\sigma_1^2$ and $\sigma_2^2$. In the light of these results, we identify several shortcomings and open questions:

*What are the lower bounds for inexact second-order methods?*
*What is the optimal trade-off between inexactness in the gradients and the Hessian?*

In this work, we attempt to answer these questions in a systematic manner. Detailed descriptions of other relevant studies can be found in Appendix A.

Table 1: Comparison of existing results for second-order methods under inexact feedback. $T$ denotes the number of iterations, and $L_2$ represents the Lipschitz constant of the Hessian.

| Algorithm | Inexactness | Gradient convergence | Hessian convergence | Exact convergence |
|---|---|---|---|---|
| Accelerated Inexact Cubic Newton (Ghadimi et al., 2017) | exact gradient $\delta_2$-inexact Hessian [1] | ✗ | $O\left(\frac{\delta_2 R^2}{T^2}\right)$ | $O\left(\frac{L_2 R^3}{T^3}\right)$ |
| Accelerated Inexact Tensor Method [2] (Agafonov et al., 2023) | $\delta_1$-inexact gradient [3] $\delta_2$-inexact Hessian [4] | $O(\delta_1 R)$ | $O\left(\frac{\delta_2 R^2}{T^2}\right)$ | $O\left(\frac{L_2 R^3}{T^3}\right)$ |
| Extra-Newton (Antonakopoulos et al., 2022) | stochastic gradient (2) unbiased stochastic Hessian [5] | $O\left(\frac{\sigma_1 R}{\sqrt{T}}\right)$ | $O\left(\frac{\sigma_2 R^2}{T^{3/2}}\right)$ | $O\left(\frac{L_2 R^3}{T^3}\right)$ |
| Accelerated Stochastic Second-order method [This Paper] | stochastic gradient (2) stochastic Hessian (4) | $O\left(\frac{\sigma_1 R}{\sqrt{T}}\right)$ | $O\left(\frac{\sigma_2 R^2}{T^2}\right)$ [6] | $O\left(\frac{L_2 R^3}{T^3}\right)$ |
| Lower bound [This Paper] | stochastic gradient (2) stochastic Hessian (4) | $\Omega\left(\frac{\sigma_1 R}{\sqrt{T}}\right)$ | $\Omega\left(\frac{\sigma_2 R^2}{T^2}\right)$ | $\Omega\left(\frac{L_2 R^3}{T^{7/2}}\right)$ |

**Contributions.** We summarize our contributions as follows:

---

[1] $\delta_2$-inexact Hessian: $\frac{\delta_2}{2} I \preceq H_x - \nabla^2 f(x) \preceq \delta_2 I$

[2] It is worth noting that the Accelerated Inexact Tensor Method can also be applied to the case of stochastic derivatives. Specifically, when $p = 2$, the total number of stochastic gradient computations is on the order of $O(\varepsilon^{-7/3})$, while the total number of stochastic Hessian computations is on the order of $O(\varepsilon^{-2/3})$ Agafonov et al. (2023). In our work, we propose an algorithm that achieves the same number of stochastic Hessian computations but significantly improves the number of stochastic gradient computations to $O(\varepsilon^{-2})$.

[3] $\delta_1$-inexact gradient: $\|g_x - \nabla f(x)\| \leq \delta_1$

[4] $\delta_2$-inexact Hessian: $\left\| \left(H_x - \nabla^2 f(x)\right)(y - x) \right\| \leq \delta_2 \|y - x\|$

[5] Unbiased stochasic Hessian: $\mathbb{E}[H(x, \xi) \mid x] = \nabla^2 f(x), \quad \mathbb{E}\left[\left\|H(x, \xi) - \nabla^2 f(x)\right\|^2 \mid x\right] \leq \sigma_2^2$

[6] Under assumption of $\delta_2$-inexact Hessian the convergence is $O\left(\frac{\delta_2 R^2}{T^2}\right)$

1. We propose an accelerated second-order algorithm that achieves the convergence rate of $O\left(\frac{\sigma_1 R}{\sqrt{T}} + \frac{\sigma_2 R^2}{T^2} + \frac{L_2 R^3}{T^3}\right)$ for stochastic Hessian with variance $\sigma_2^2$ and $O\left(\frac{\sigma_1 R}{\sqrt{T}} + \frac{\delta_2 R^2}{T^2} + \frac{L_2 R^3}{T^3}\right)$ for $\delta_2$-inexact Hessian, improving the existing results (Agafonov et al., 2023; Antonakopoulos et al., 2022) (see Table 1).

2. We prove that the above bounds are tight with respect to the variance of the gradient and the Hessian by developing a matching theoretical complexity lower bound (see Table 1).

3. Our algorithm involves solving a cubic subproblem that arises in several globally convergent second-order methods (Nesterov & Polyak, 2006; Nesterov, 2008). To address this, we propose a criterion based on the accuracy of the subproblem's gradient, along with a dynamic strategy for selecting the appropriate level of inexactness. This ensures an efficient solution of the subproblems without sacrificing the fast convergence of the initial method.

4. We extend our method for higher-order minimization with stochastic/inexact oracles. We achieve the $O\left(\frac{\sigma_1 R}{\sqrt{T}} + \sum_{i=2}^{p} \frac{\delta_i R^i}{T^i} + \frac{L_p R^{p+1}}{T^{p+1}}\right)$ rate with $\delta_i$-inexact $i$-th derivative.

5. We propose a restarted version of our algorithm for strongly convex minimization, which exhibits a linear rate. Via a mini-batch strategy, we demonstrate that the total number of Hessian computations scales linearly with the desired accuracy $\varepsilon$.

## 2 PROBLEM STATEMENT AND PRELIMINARIES

**Taylor approximation and oracle feedback.** Our starting point for constructing second-order method is based primarily on the second-order Taylor approximation of the function $f(x)$

$$\Phi_x(y) \stackrel{\text{def}}{=} f(x) + \langle \nabla f(x), y - x \rangle + \frac{1}{2} \langle y - x, \nabla^2 f(x)(y - x) \rangle, \quad y \in \mathbb{R}^d.$$

In particular, since the exact computation of the Hessians can be a quite tiresome task, we attempt to employ more tractable inexact estimators $g(x)$ and $H(x)$ for the gradient and Hessian. These estimators are going to be the main building blocks for the construction of the "inexact" second-order Taylor approximation. Formally, this is given by:

$$\phi_x(y) = f(x) + \langle g(x), y - x \rangle + \frac{1}{2} \langle y - x, H(x)(y - x) \rangle, \quad y \in \mathbb{R}^d. \tag{3}$$

Therefore, by combining Assumption 1.1 with the aforementioned estimators, we readily get the following estimation:

**Lemma 2.1** ((Agafonov et al., 2023, Lemma 2)). *Let Assumption 1.1 hold. Then, for any $x, y \in \mathbb{R}^d$, we have*

$$|f(y) - \phi_x(y)| \leq \left(\|g(x) - \nabla f(x)\| + \frac{1}{2}\|\left(H(x) - \nabla^2 f(x)\right)(y - x)\|\right)\|y - x\| + \frac{L_2}{6}\|y - x\|^3.$$

$$\|\nabla f(y) - \nabla \phi_x(y)\| \leq \|g(x) - \nabla f(x)\| + \|\left(H(x) - \nabla^2 f(x)\right)(y - x)\| + \frac{L_2}{3}\|y - x\|^2.$$

Now, having established the main toolkit concerning the approximation of $f$ in the rest of this section, we introduce the blanket assumptions regarding the inexact gradients and Hessians (for a complete overview, we refer to Table 1). In particular, we assume that our estimators satisfy the following statistical conditions.

**Assumption 2.2** (Unbiased stochastic gradient with bounded variance and stochastic Hessian with bounded variance). *For all $x \in \mathbb{R}^d$, stochastic gradient $g(x, \xi)$ satisfies (2) and stochastic Hessian $H(x, \xi)$ satisfies*

$$\mathbb{E}\left[\|H(x, \xi) - \nabla^2 f(x)\|_2^2 \mid x\right] \leq \sigma_2^2. \tag{4}$$

**Assumption 2.3** (Unbiased stochastic gradient with bounded variance and inexact Hessian). *For all $x \in \mathbb{R}^d$ stochastic gradient $g(x, \xi)$ satisfies (2). For given $x$, $y \in \mathbb{R}^d$ inexact Hessian $H(x)$ satisfies*

$$\|(H(x) - \nabla^2 f(x))[y - x]\| \leq \delta_2^{x,y}\|y - x\|. \tag{5}$$

Assumptions 2.2 and 2.3 differ from Condition 1 in (Agafonov et al., 2023) by the unbiasedness of the gradient. An unbiased gradient allows us to attain optimal convergence in the corresponding term $O(1/\sqrt{T})$, while an inexact gradient slows down the convergence to $O(1)$ since a constant error can misalign the gradient. Note, that we do not assume the unbiasedness of the Hessian in all assumptions. Finally, note that Assumption 2.3 does not require (5) to be met for all $x$, $y \in \mathbb{R}^d$. Instead, we only consider inexactness along the direction $y - x$, which may be significantly less than the norm of the difference between Hessian and its approximation $H(x)$.

**Auxiliary problem.** Most second-order methods with global convergence require solving an auxiliary subproblem at each iteration. However, to the best of our knowledge, existing works that consider

convex second-order methods under inexact derivatives do not account for inexactness in the solution of the subproblem. To address this gap, we propose incorporating a gradient criteria for the subproblem solution, given by

$$\min_{y \in \mathbb{R}^d} \omega_x(y) \text{ such that } \|\nabla \omega_x(y)\| \leq \tau, \tag{6}$$

where $\omega_x(y)$ is the objective of subproblem and $\tau \geq 0$ is a tolerance parameter. We highlight that this criterion is verifiable at each step of the algorithm, which facilitates determining when to stop. By setting a constant tolerance parameter $\tau$, we get the following relationship between the absolute accuracy $\epsilon$ required for the initial problem and $\tau$: $\tau = O\left(\epsilon^{\frac{5}{6}}\right)$. In practice, it may not be necessary to use a very small accuracy in the beginning. Later, we will discuss strategies for choosing the sequence of $\tau_t$ based on the number of iterations $t$.

## 3 THE METHOD

In this section, we present our proposed method, dubbed as Accelerated Stochastic Cubic Regularized Newton's method. In particular, extending on recent accelerated second-order algorithms (Nesterov, 2021b; Ghadimi et al., 2017; Agafonov et al., 2023), we propose a new variant of the accelerated cubic regularization method with stochastic gradients that achieves optimal convergence in terms corresponding to gradient and Hessian inexactness. Moreover, the proposed scheme allows for the approximate solution of the auxiliary subproblem, enabling a precise determination of the required level of subproblem accuracy.

We begin the algorithm description by introducing the main step. Given constants $\bar{\delta} > 0$ and $M \geq L_2$, we define a model of the objective

$$\omega_x^{M,\bar{\delta}}(y) := \phi_x(y) + \frac{\bar{\delta}}{2}\|x - y\|^2 + \frac{M}{6}\|x - y\|^3.$$

At each step of the algorithm, we aim to find $u \in \arg\min_{y \in \mathbb{R}^d} \omega_x^{M,\bar{\delta}}(y)$. However, finding the respective minimizer is a separate challenge. Instead of computing the exact minimum, we aim to find a point $s \in \mathbb{R}^d$ with a small norm of the gradient.

**Definition 3.1.** *Denote by $s^{M,\bar{\delta},\tau}(x)$ a $\tau$-inexact solution of subproblem, i.e. a point $s := s^{M,\bar{\delta},\tau}(x)$ such that*

$$\|\nabla \omega_x^{M,\bar{\delta}}(s)\| \leq \tau.$$

Next, we employ the technique of estimating sequences to propose the Accelerated Stochastic Cubic Newton method. Such acceleration is based on aggregating stochastic linear models given by

$$l(x, y) = f(y) + \langle g(y, \xi), x - y \rangle$$

in function $\psi_t(x)$ (8), (9). The method is presented in detail in Algorithm 1.

---

**Algorithm 1** Accelerated Stochastic Cubic Newton

---

1: **Input:** $y_0 = x_0$ is starting point; constants $M \geq 2L_2$; non-negative non-decreasing sequences $\{\bar{\delta}_t\}_{t\geq 0}, \{\lambda_t\}_{t\geq 0}, \{\bar{\kappa}_2^t\}_{t\geq 0}, \{\bar{\kappa}_3^t\}_{t\geq 0}$, and

$$\alpha_t = \frac{3}{t+3}, \quad A_t = \prod_{j=1}^{t}(1 - \alpha_j), \quad A_0 = 1, \tag{7}$$

$$\psi_0(x) := \frac{\bar{\kappa}_2^0 + \lambda_0}{2}\|x - x_0\|^2 + \frac{\bar{\kappa}_3^0}{3}\|x - x_0\|^3. \tag{8}$$

2: **for** $t \geq 0$ **do**
3:

$$v_t = (1 - \alpha_t)x_t + \alpha_t y_t, \quad x_{t+1} = s^{M,\bar{\delta}_t,\tau}(v_t)$$

4:     Compute

$$y_{t+1} = \arg\min_{x \in \mathbb{R}^n} \left\{ \psi_{t+1}(x) := \psi_t(x) + \frac{\lambda_{t+1} - \lambda_t}{2}\|x - x_0\|^2 \right.$$

$$\left. + \sum_{i=2}^{3} \frac{\bar{\kappa}_i^{t+1} - \bar{\kappa}_i^t}{i}\|x - x_0\|^i + \frac{\alpha_t}{A_t}l(x, x_{t+1}) \right\}. \tag{9}$$

---

**Theorem 3.2.** *Let Assumption 1.1 hold and $M \geq 2L_2$.*

- *Let Assumption 2.2 hold. After $T \geq 1$ with parameters*

$$\bar{\kappa}_2^{t+1} = \frac{2\bar{\delta}_t \alpha_t^2}{A_t}, \ \bar{\kappa}_3^{t+1} = \frac{8M}{3}\frac{\alpha_{t+1}^3}{A_{t+1}}, \ \lambda_t = \frac{\sigma_1}{R}(t+3)^{\frac{5}{2}}, \ \bar{\delta}_t = 2\sigma_2 + \frac{\sigma_1 + \tau}{R}(t+3)^{\frac{3}{2}}, \tag{10}$$

*we get the following bound*

$$\mathbb{E}\left[f(x_T) - f(x^*)\right] \leq O\left(\frac{\tau R}{\sqrt{T}} + \frac{\sigma_1 R}{\sqrt{T}} + \frac{\sigma_2 R^2}{T^2} + \frac{MR^3}{T^3}\right). \tag{11}$$

- *Let Assumption 2.3 hold. After $T \geq 1$ with parameters defined in* (10) *and* $\sigma_2 = \delta_2 = \max_{t=1,\ldots,T} \delta_t^{v_{t-1},x_t}$, *we get the following bound*

$$\mathbb{E}[f(x_T) - f(x^*)] \leq O\left(\frac{\tau R}{\sqrt{T}} + \frac{\sigma_1 R}{\sqrt{T}} + \frac{\delta_2 R^2}{T^2} + \frac{MR^3}{T^3}\right). \tag{12}$$

This result provides an upper bound for the objective residual after $T$ iterations of Algorithm 1. The last term in the RHS of (11) and (12) corresponds to the case of exact Accelerated Cubic Newton method (Nesterov, 2008). The remaining terms reveal how the convergence rate is affected by the imprecise calculation of each derivative and by inexact solution of subproblem. We provide sufficient conditions for the inexactness in the derivatives to ensure that the method can still obtain an objective residual smaller than $\varepsilon$. Specifically, this result addresses the following question: given that the errors are controllable and can be made arbitrarily small, how small should each derivative's error be to achieve an $\varepsilon$-solution?

**Corollary 3.3.** *Let assumptions of Theorem 3.2 hold and let $\varepsilon > 0$ be the desired solution accuracy.*

- *Let the levels of inexactness in Assumption 2.2 be:*

$$\tau = O\left(\varepsilon^{\frac{5}{6}}\left(\frac{M}{R^3}\right)^{\frac{1}{6}}\right), \quad \sigma_1 = O\left(\varepsilon^{\frac{5}{6}}\left(\frac{M}{R^3}\right)^{\frac{1}{6}}\right), \quad \sigma_2 = O\left(\varepsilon^{\frac{1}{3}} M^{\frac{2}{3}}\right)$$

- *Let the levels of inexactness in Assumption 2.3 be:*

$$\tau = O\left(\varepsilon^{\frac{5}{6}}\left(\frac{M}{R^3}\right)^{\frac{1}{6}}\right), \quad \sigma_1 = O\left(\varepsilon^{\frac{5}{6}}\left(\frac{M}{R^3}\right)^{\frac{1}{6}}\right), \quad \delta_2 = O\left(\varepsilon^{\frac{1}{3}} M^{\frac{2}{3}}\right)$$

*And let the number of iterations of Algorithm 1 satisfy $T = O\left(\left(\frac{MR^3}{\varepsilon}\right)^{\frac{1}{3}}\right)$. Then $x_T$ is an $\varepsilon$-solution of problem* (1), *i.e.* $f(x_T) - f(x^*) \leq \varepsilon$.

In practice, achieving an excessively accurate solution for the subproblem on the initial iterations is not essential. Instead, a dynamic strategy can be employed to determine the level of accuracy required for the subproblem. Specifically, we can choose a dynamic precision level according to $\tau_t = \frac{c}{t^{5/2}}$, where $c > 0$. As a result, the convergence rate term associated with the inexactness of the subproblem becomes $O\left(\frac{c}{T^3}\right)$, which matches the convergence rate of the Accelerated Cubic Newton method.

## 4 THEORETICAL COMPLEXITY LOWER BOUND

In this section, we present a novel theoretical complexity lower bound for inexact second-order methods with stochastic gradient and inexact (stochastic) Hessian. The proof technique draws inspiration from the works (Devolder et al., 2014; Nesterov, 2021b; 2018). For this section, we assume that the function $f(x)$ is convex and has $L_1$-Lipschitz-continuous gradient and $L_2$-Lipschitz-continuous Hessian.

To begin, we describe the information and structure of stochastic second-order methods. At each point $x_t$, the oracle provides us with an unbiased stochastic gradient $g_t = g(x_t, \xi)$ and an inexact (stochastic) Hessian $H_t = H(x_t, \xi)$. The method can compute the minimum of the following models:

$$h_{t+1} = \operatorname{argmin}_h \left\{\phi_{x_t}(h) = a_1 \langle g_t, h \rangle + a_2 \langle H_t h, h \rangle + b_1 \|h\|^2 + b_2 \|h\|^3\right\}.$$

Now, we formulate the main assumption regarding the method's ability to generate new points.

**Assumption 4.1.** *The method generates a recursive sequence of test points $x_t$ that satisfies the following condition*

$$x_{t+1} \in x_0 + \operatorname{Span}\{h_1, \ldots, h_{t+1}\}$$

Most first-order and second-order methods, including accelerated versions, typically satisfy this assumption. However, we highlight that randomized methods are not covered by this lower bound. Randomized lower bound even for exact high-order methods is still an open problem. More details on randomized lower bounds for first-order methods are presented in (Woodworth & Srebro, 2017; Nemirovski & Yudin, 1983). Finally, we present the main theoretical complexity lower bound theorem for stochastic second-order methods.

**Theorem 4.2.** *Let some second-order method $\mathcal{M}$ with exactly solved subproblem satisfy Assumption 4.1 and have access only to unbiased stochastic gradient and inexact Hessian satisfying Assumption 2.2 or Assumption 2.3 with $\sigma_2 = \delta_2 = \max_{t=1,\ldots,T} \delta_t^{x_{t-1},x_t}$. Assume the method $\mathcal{M}$ ensures for any function $f$ with $L_1$-Lipschitz-continuous gradient and $L_2$-Lipschitz-continuous Hessian the following convergence rate*

$$\min_{0 \leq t \leq T} \mathbb{E}\left[f(x_t) - f(x^*)\right] \leq O(1) \max\left\{\frac{\sigma_1 R}{\Xi_1(T)}; \frac{\sigma_2 R^2}{\Xi_2(T)}; \frac{L_2 R^3}{\Xi_3(T)}\right\}. \tag{13}$$

*Then for all $T \geq 1$ we haven*
$$\Xi_1(T) \leq \sqrt{T}, \qquad \Xi_2(T) \leq T^2, \qquad \Xi_3(T) \leq T^{7/2}. \qquad (14)$$

*Proof.* We prove this Theorem from contradiction. Let assume that there exist the method $\mathcal{M}$ that satisfies conditions of the Theorem 4.2 and it is faster in one of the bounds from (14).

The first case, $\Xi_1(T) > \sqrt{T}$ or $\Xi_2(T) > T^2$. Let us apply this method for the first-order lower bound function. It is well-known, that for the first-order methods, the lower bound is $\Omega\left(\frac{\sigma_1 R}{\sqrt{T}} + \frac{L_1 R^2}{T^2}\right)$ (Nemirovski & Yudin, 1983). Also, the first-order lower bound function has 0-Lipschitz-continuous Hessian. It means, that the method $\mathcal{M}$ can be applied for the first-order lower-bound function. We fix stochastic Hessian oracle as $H(x, \xi) = 2L_1 I$. It means that $\sigma_2 = 2L$ for such inexact Hessian. With such matrix $H(x, \xi) = 2L_1 I$, the method $\mathcal{M}$ has only the first-order information and lies in the class of first-order methods. Hence, we apply the method $\mathcal{M}$ to the first-order lower bound function and get the rate $\min_{0 \leq t \leq T} \mathbb{E}\left[f(x_t) - f(x^*)\right] \leq O(1) \max\left\{\frac{\sigma_1 R}{\Xi_1(T)}; \frac{\sigma_2 R^2}{\Xi_2(T)}\right\}$, where $\Xi_1(T) > \sqrt{T}$ or $\Xi_2(T) > T^2$. It means that we've got a faster method than a lower bound. It is a contradiction, hence the rates for the method $\mathcal{M}$ are bounded as $\Xi_1(T) \leq \sqrt{T}, \Xi_2(T) \leq T^2$. The second case, $\Xi_3(T) > T^{7/2}$. It is well-known, that the deterministic second-order lower bound is $\Omega\left(\frac{L_2 R^3}{T^{7/2}}\right)$. Let us apply the method $\mathcal{M}$ for the second-order lower bound function, where the oracle give us exact gradients and exact Hessians, then $\sigma_1 = 0$, $\sigma_2 = 0$ and the method $\mathcal{M}$ is in class of exact second-order methods but converges faster than the lower bound. It is a contradiction, hence the rate for the method $\mathcal{M}$ is bounded as $\Xi_3(T) \leq T^{7/2}$. $\qquad \square$

## 5 TENSOR GENERALIZATION

In this section we propose a tensor generalization of Algorithm 1. We start with introducing the standard assumption on the objective $f$ for tensor methods.

**Assumption 5.1.** *Function $f$ is convex, $p$ times differentiable on $\mathbb{R}^d$, and its $p$-th derivative is Lipschitz continuous, i.e. for all $x, y \in \mathbb{R}^d$*
$$\|\nabla^p f(x) - \nabla^p f(y)\| \leq L_p \|x - y\|.$$

We denote the $i$-th directional derivative of function $f$ at $x$ along directions $s_1, \ldots, s_i \in \mathbb{R}^n$ as $\nabla^i f(x)[s_1, \ldots, s_i]$. If all directions are the same we write $\nabla^i f(x)[s]^i$. For a $p$-th order tensor $U$, we denote by $\|U\|$ its tensor norm recursively induced (Cartis et al., 2017) by the Euclidean norm on the space of $p$-th order tensors:
$$\|U\| = \max_{\|s_1\| = \ldots = \|s_p\| = 1}\{|U[s_1, \ldots, s_p]|\},$$
where $\|\cdot\|$ is the standard Euclidean norm.
We construct tensor methods based on the $p$-th order Taylor approximation of the function $f(x)$, which can be written as follows:
$$\Phi_{x,p}(y) \overset{\text{def}}{=} f(x) + \sum_{i=1}^p \frac{1}{i!} \nabla^i f(x)[y - x]^i, \quad y \in \mathbb{R}^d.$$
Using approximations $G_i(x)$ for the derivatives $\nabla^i f(x)$ we create an inexact $p$-th order Taylor series expansion of the objective
$$\phi_{x,p}(y) = f(x) + \sum_{i=1}^p \frac{1}{i!} G_i(x)[y - x]^i.$$
Next, we introduce a counterpart of Lemma 2.1 for high-order methods.

**Lemma 5.2** ((Agafonov et al., 2023, Lemma 2)). *Let Assumption 5.1 hold. Then, for any $x, y \in \mathbb{R}^d$, we have*
$$|f(y) - \phi_{x,p}(y)| \leq \sum_{i=1}^p \frac{1}{i!} \|(G_i(x) - \nabla^i f(x))[y - x]^{i-1}\| \|y - x\| + \frac{L_p}{(p+1)!} \|y - x\|^{p+1},$$
$$\|\nabla f(y) - \nabla \phi_{x,p}(y)\| \leq \sum_{i=1}^p \frac{1}{(i-1)!} \|(G_i(x) - \nabla^i f(x))[y - x]^{i-1}\| + \frac{L_p}{p!} \|y - x\|^p,$$

*where we use the standard convention $0! = 1$.*

Following the assumptions for the second-order method we introduce analogical assumptions for high-order method.

**Assumption 5.3** (Unbiased stochastic gradient with bounded variance and stochastic high-order derivatives with bounded variance). *For any $x \in \mathbb{R}^d$ stochastic gradient $G_1(x, \xi)$ and stochastic*

high-order derivatives $G_i(x, \xi)$, $i = 2, \ldots, p$ satisfy

$$\mathbb{E}[G_1(x, \xi) \mid x] = \nabla f(x), \quad \mathbb{E}\left[\|G_1(x, \xi) - \nabla f(x)\|^2 \mid x\right] \leq \sigma_1^2, \tag{15}$$

$$\mathbb{E}\left[\|G_i(x, \xi) - \nabla^i f(x)\|^2 \mid x\right] \leq \sigma_i^2, \ i = 2, \ldots, p.$$

**Assumption 5.4** (Unbiased stochastic gradient with bounded variance and inexact high-order derivatives). *For any $x \in \mathbb{R}^d$ stochastic gradient $G_1(x, \xi)$ satisfy* (15). *For given $x, \ y \in \mathbb{R}^d$ inexact high-order derivatives $G_i(x)$, $i = 2, \ldots, p$ satisfy*

$$\|(G_i(x) - \nabla^i f(x))[y - x]^{i-1}\| \leq \delta_i^{x,y} \|y - x\|^{i-1}.$$

To extend Algorithm 1 to tensor methods, we introduce a $p$-th order model of the function:

$$\omega_{x,p}^{M,\bar{\delta}}(y) := \phi_{x,p}(y) + \tfrac{\bar{\delta}}{2}\|x - y\|^2 + \sum_{i=3}^{p} \tfrac{\eta_i \delta_i}{i!}\|x - y\|^i + \tfrac{pM}{(p+1)!}\|x - y\|^{p+1},$$

where $\eta_i > 0$, $3 \leq i \leq p$. Next, we modify Definition 3.1 for the high order derivatives case

**Definition 5.5.** *Denote by $S_p^{M,\bar{\delta},\tau}(x)$ a point $S := S_p^{M,\bar{\delta},\tau}(x)$ such that $\|\nabla \omega_{x,p}^{M,\bar{\delta}}(S)\| \leq \tau$.*
Now, we are prepared to introduce the method and state the convergence theorem.

---

**Algorithm 2** Accelerated Stochastic Tensor Method

---

1: **Input:** $y_0 = x_0$ is starting point; constants $M \geq \tfrac{2}{p}L_p$; $\eta_i \geq 4$, $3 \leq i \leq p$; starting inexactness $\bar{\delta}_0 \geq 0$; nonnegative nondecreasing sequences $\{\bar{\kappa}_i^t\}_{t \geq 0}$ for $i = 2, \ldots, p + 1$, and

$$\alpha_t = \tfrac{p+1}{t+p+1}, \quad A_t = \prod_{j=1}^{t}(1 - \alpha_j), \quad A_0 = 1. \tag{16}$$

$$\psi_0(x) := \tfrac{\bar{\kappa}_2^0 + \lambda_0}{2}\|x - x_0\|^2 + \sum_{i=3}^{p} \tfrac{\bar{\kappa}_i^0}{i!}\|x - x_0\|^i.$$

2: **for** $t \geq 0$ **do**
3:

$$v_t = (1 - \alpha_t)x_t + \alpha_t y_t, \quad x_{t+1} = S_p^{M,\bar{\delta}_t,\tau}(v_t)$$

4: Compute

$$y_{t+1} = \arg\min_{x \in \mathbb{R}^n} \left\{ \psi_{t+1}(x) := \psi_t(x) + \tfrac{\lambda_{t+1} - \lambda_t}{2}\|x - x_0\|^2 \right.$$
$$\left. + \sum_{i=2}^{p} \tfrac{\bar{\kappa}_i^{t+1} - \bar{\kappa}_i^t}{i!}\|x - x_0\|^i + \tfrac{\alpha_t}{A_t}l(x, x_{t+1}) \right\}.$$

---

**Theorem 5.6.** *Let Assumption 5.1 hold and $M \geq \tfrac{2}{p}L_p$.*

- *Let Assumption 5.3 hold. After $T \geq 1$ with parameters*

$$\bar{\kappa}_2^t = O\left(\tfrac{\bar{\delta}_t \alpha_t^2}{A_t}\right), \ \bar{\kappa}_i^{t+1} = O\left(\tfrac{\alpha_{t+1}^i \delta_i}{A_{t+1}}\right), \ \bar{\kappa}_{p+1}^{t+1} = O\left(\tfrac{\alpha_{t+1}^{p+1} M}{A_{t+1}}\right),$$

$$\lambda_t = O\left(\tfrac{\sigma_1}{R}t^{p+1/2}\right), \ \delta_t = O\left(\sigma_2 + \tfrac{\sigma_1 + \tau}{R}t^{\frac{3}{2}}\right) \tag{17}$$

  *we get the following bound*

$$\mathbb{E}[f(x_T) - f(x^*)] \leq O\left(\tfrac{\tau R}{\sqrt{T}} + \tfrac{\sigma_1 R}{\sqrt{T}} + \sum_{i=2}^{p} \tfrac{\sigma_i R^i}{T^i} + \tfrac{MR^{p+1}}{T^{p+1}}\right).$$

- *Let Assumption 5.4 hold. After $T \geq 1$ with parameters defined in* (10) *and $\sigma_i = \delta_i = \max_{t=1,\ldots,T} \delta_{i,t}^{v_{t-1}, x_t}$ we get the following bound*

$$\mathbb{E}[f(x_T) - f(x^*)] \leq O\left(\tfrac{\tau R}{\sqrt{T}} + \tfrac{\sigma_1 R}{\sqrt{T}} + \sum_{i=2}^{p} \tfrac{\delta_i R^i}{T^i} + \tfrac{MR^{p+1}}{T^{p+1}}\right).$$

## 6 STRONGLY CONVEX CASE

**Assumption 6.1.** *Function $f$ is $\mu$-strongly convex, $p$ times differentiable on $\mathbb{R}^d$, and its $p$-th derivative is Lipschitz continuous, i.e. for all $x, y \in \mathbb{R}^d$*

$$\|\nabla^p f(x) - \nabla^p f(y)\| \leq L_p \|x - y\|.$$

To exploit the strong convexity of the objective function and attain a linear convergence rate, we introduce a restarted version of Restarted Accelerated Stochastic Tensor Method (Algorithm 2). In each iteration of Restarted Accelerated Stochastic Tensor Method (Algorithm 3), we execute Algorithm 2 for a predetermined number of iterations as specified in equation (18). The output of this run is then used as the initial point for the subsequent iteration of Algorithm 1, which resets the parameters, and this process repeats iteratively.

---

**Algorithm 3** Restarted Accelerated Stochastic Tensor Method

---

**Input**: $z_0 \in \mathbb{R}^d$, strong convexity parameter $\mu > 0$, $M \geq L_p$, and $R_0 > 0$ such that $\|z_0 - x^*\| \leq R_0$.

**For** $s = 1, 2, \ldots$**:**

1. Set $x_0 = z_{s-1}$, $r_{s-1} = \frac{R_0}{2^{s-1}}$, and $R_{s-1} = \|z_{s-1} - x^*\|$.

2. Run Algorithm 2 for $t_s$ iterations, where

$$t_s = O(1)\max\left\{1, \left(\frac{\tau}{\mu r_{s-1}}\right)^2, \left(\frac{\sigma_1}{\mu r_{s-1}}\right)^2, \max_{i=2,\ldots,p}\left(\frac{\delta_i R_{s-1}^{i-2}}{\mu}\right)^{\frac{1}{i}}, \left(\frac{L_p R_{s-1}^{p-1}}{\mu}\right)^{\frac{1}{p+1}}\right\}. \quad (18)$$

3. Set $z_s = x^{t_s}$.

---

**Theorem 6.2.** *Let Assumption 6.1 hold and let parameters of Algorithm 1 be chosen as in (17). Let $\{z_s\}_{s\geq 0}$ be generated by Algorithm 3 and $R > 0$ be such that $\|z_0 - x^*\| \leq R$. Then for any $s \geq 0$ we have*

$$\mathbb{E}\|z_s - x^*\|^2 \leq 4^{-s}R^2, \qquad\qquad \mathbb{E}f(z_s) - f(x^*) \leq 2^{-2s-1}\mu R^2. \quad (19)$$

*Moreover, the total number of iterations to reach desired accuracy $\varepsilon$ : $f(z_s) - f(x^*) \leq \varepsilon$ in expectation is*

$$O\left(\frac{(\tau+\sigma_1)^2}{\mu\varepsilon} + \left(\sqrt{\frac{\sigma_2}{\mu}} + 1\right)\log\frac{f(z_0)-f(x^*)}{\varepsilon} + \sum_{i=3}^p\left(\frac{\sigma_i R^{i-2}}{\mu}\right)^{\frac{1}{i}} + \left(\frac{L_p R^{p-1}}{\mu}\right)^{\frac{1}{p+1}}\right).$$

Now, let us make a few observations regarding the results obtained in Theorem 6.2. For simplicity let solution of the subproblem be exact and $p = 2$, i.e. we do the restarts of the Accelerated Stochastic Cubic Newton, so the total number of iterations is

$$O\left(\frac{\sigma_1^2}{\mu\varepsilon} + \left(\sqrt{\frac{\sigma_2}{\mu}} + 1\right)\log\frac{f(z_0)-f(x^*)}{\varepsilon} + \left(\frac{L_2 R}{\mu}\right)^{\frac{1}{3}}\right). \quad (20)$$

Next, let's consider solving the stochastic optimization problem $\min_{x\in\mathbb{R}^d} F(x) = \mathbb{E}[f(x,\xi)]$ using the mini-batch Restarted Accelerated Stochastic Cubic Newton method (Algorithm 3) with $p = 2$. In this approach, the mini-batched stochastic gradient is computed as $\frac{1}{r_1}\sum_{i=1}^{r_1}\nabla f(x,\xi_i)$ and the mini-batched stochastic Hessian is computed as $\frac{1}{r_2}\sum_{i=1}^{r_2}\nabla^2 f(x,\xi_i)$, where $r_1$ and $r_2$ represent the batch sizes for gradients and Hessians, respectively.

From the convergence estimates in (19) and (20), we can determine the required sample sizes for computing the batched gradients and batched Hessians. Specifically, we have $r_1 = \tilde{O}\left(\frac{\sigma_1^2}{\varepsilon\mu^{2/3}}\right)$ and $r_2 = O\left(\frac{\sigma_2}{\mu^{1/3}}\right)$. Consequently, the overall number of stochastic gradient computations is $O\left(\frac{\sigma_1^2}{\varepsilon\mu^{2/3}}\right)$, which is similar to the accelerated SGD method (Ghadimi & Lan, 2013). Interestingly, the number of stochastic Hessian computations scales linearly with the desired accuracy $\varepsilon$, i.e., $O\left(\frac{\sigma_2}{\mu^{1/3}}\log\frac{1}{\varepsilon}\right)$.

This result highlights the practical importance of second-order methods. Since the batch size of the Hessian is constant, there is no need to adjust it as the desired solution as accuracy increases. This is particularly useful in distributed optimization problems under the assumption of beta similarity (Zhang & Lin, 2015). In methods with such assumption (Zhang & Lin, 2015; Daneshmand et al., 2021; Agafonov et al., 2023), the server stores a Hessian sample that provides a "good" approximation of the exact Hessian of the objective function. Algorithms utilize this approximation instead of exchanging curvature information with the workers. The constant batch size allows for accurately determining the necessary sample size to achieve fast convergence to any desired accuracy.

## 7 EXPERIMENTS

In this section, we present numerical experiments conducted to demonstrate the efficiency of our proposed methods. We consider logistic regression problems of the form:

$$f(x) = \mathbb{E}\left[\log(1 + \exp(-b_\xi \cdot a_\xi^\top x))\right],$$

where $(a_\xi, b_\xi)$ are the training samples described by features $a_\xi \in \mathbb{R}^d$ and class labels $b_i \in \{-1, 1\}$.

**Setup.** We present results on the a9a dataset ($d = 123$) from LibSVM by Chang & Lin (2011). We demonstrate the performance of Accelerated Stochastic Cubic Newton in three regimes: deterministic oracles (Figure 1), stochastic oracles with the same batch size for gradient and Hessians (Figures 2a, 2b), and stochastic oracles with smaller batch size for Hessians (Figures 2c, 2d).

The final mode is especially intriguing because the convergence component of Algorithm 1 associated with gradient noise decreases as $1/\sqrt{t}$, while the component related to Hessian noise decreases as $1/t^2$. This enables the use of smaller Hessian batch sizes (see Corollary 3.3).

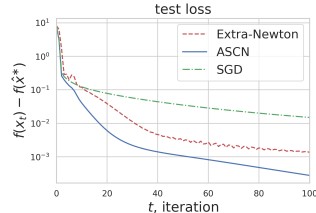

For stochastic experiments, we randomly split the dataset into training (30000 data samples) and test (2561 data samples) sets. The methods randomly sample data from the training set and do not have access to the test data. In this case, the training loss represents finite sum minimization properties, and the test loss represents expectation minimization. We compare the performance of the SGD, Extra-Newton (EN), and Accelerated Stochastic Cubic Newton (ASCN). We present experiments for fine-tuned hyperparameters in Figures 1, 2. For SGD, we've fine-tuned 1 parameter $lr$. For EN, we've fine-tuned 2 parameters: $\gamma$ and $\beta_0$. For ASCN, we've fine-tuned 2 parameters: $M$ and $\frac{\sigma_1}{R}$ (only for stochastic case) as the entity, also $\tau = 0$ and $\sigma_2 = 0$ as they are dominated by $\frac{\sigma_1}{R}$. To demonstrate the globalization properties of the methods, we consider the starting

Figure 1: Logistic regression on `a9a` with deterministic oracles

point $x_0$ far from the solution, specifically $x_0 = 3 \cdot e$, where $e$ is the all-one vector. All methods are implemented as PyTorch 2.0 optimizers. Additional details and experiments are provided in the Appendix C.

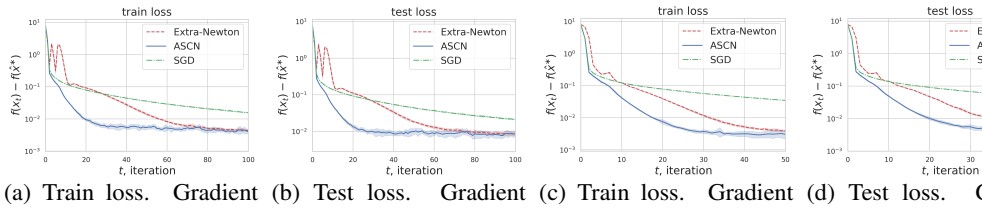

(a) Train loss. Gradient and Hessian batch sizes are 1500

(b) Test loss. Gradient and Hessian batch sizes are 1500

(c) Train loss. Gradient batch size is 10000, Hessian batch size is 150

(d) Test loss. Gradient batch size is 10000, Hessian batch size is 150

Figure 2: Logistic regression on `a9a` with stochastic oracles

**Results.** The ASCN method proposed in this study consistently outperforms Extra-Newton and SGD across all experimental scenarios. In deterministic settings, ASCN exhibits a slight superiority over Extra-Newton. In stochastic experiments, we observe a notable improvement as well. However, it's worth noting that in stochastic regime as we approach convergence, all methods tend to converge to the same point. This convergence pattern is primarily influenced by the stochastic gradient noise $\frac{\sigma_1 R}{\sqrt{T}}$ term, which dominates in rates as we converge to solution. Furthermore, experiment with different batch sizes for gradients and Hessians support the theory, confirming that the Hessian inexactness term in ASCN $\frac{\sigma_2 R^2}{T^2}$ has faster rate than the corresponding term in Extra-Newton $\frac{\sigma_2 R^2}{T^{3/2}}$. To conclude, the experiments show that second-order information could significantly accelerate the convergence. Moreover, the methods need significantly less stochastic Hessians than stochastic gradients.

## 8 CONCLUSION

In summary, our contribution includes a novel stochastic accelerated second-order algorithm for convex and strongly convex optimization. We establish a lower bound for stochastic second-order optimization and prove our algorithm's achievement of optimal convergence in both gradient and Hessian inexactness. Additionally, we introduce a tensor generalization of second-order methods for stochastic high-order derivatives. Nevertheless, it's essential to acknowledge certain limitations. Like other globally convergent second-order methods, our algorithm involves a subproblem that necessitates an additional subroutine to find its solution. To mitigate this challenge, we offer theoretical insights into the required accuracy of the subproblem's solution. Future research could involve enhancing the adaptiveness of the algorithm. Additionally, there is a potential for constructing optimal stochastic second-order and tensor methods by incorporating stochastic elements into existing exact methods. These efforts could further improve both practical and theoretical aspects of stochastic second-order and high-order optimization.

**Acknowledgments.** This work was supported by a grant for research centers in the field of artificial intelligence, provided by the Analytical Center for the Government of the Russian Federation in accordance with the subsidy agreement (agreement identifier 000000D730321P5Q0002) and the agreement with the Moscow Institute of Physics and Technology dated November 1, 2021 No. 70-2021-00138. This work was supported by Hasler Foundation Program: Hasler Responsible AI (project number 21043). This work was supported by the Swiss National Science Foundation (SNSF) under grant number 200021_205011. Research was sponsored by the Army Research Office and was accomplished under Grant Number W911NF-24-1-0048.

**Ethics Statement.** The authors acknowledge that they have read and adhere to the ICLR Code of Ethics.

**Reproducibility Statement.** The experimental details are provided in Section 7 and Appendix C. The PyTorch code for the methods is available on `https://github.com/OPTAMI/OPTAMI`.

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

# A   RELATED WORK

The idea of using high-order derivatives in optimization has been known for a long time Hoffmann & Kornstaedt (1978). In 2009, M. Baes extended the cubic regularization approach with second-order derivatives ($p = 2$) from Nesterov & Polyak (2006) to high-order derivatives ($p > 2$) in Baes (2009). However, the subproblem of these methods was non-convex, making them impractical. In 2018, Yu. Nesterov proposed the implementable (Accelerated) Tensor Method Nesterov (2021b), wherein the convexity of the subproblem was reached by increasing a regularization parameter. Hence, the convex subproblem could be efficiently solved by appropriate subsolvers, making the algorithm practically applicable. In the same work, a lower complexity bound for tensor methods under higher-order smoothness assumption was proposed. Shortly after, near-optimal Gasnikov et al. (2019b;a); Bubeck et al. (2019); Jiang et al. (2019) and optimal Kovalev & Gasnikov (2022); Carmon et al. (2022) high-order methods were introduced. Furthermore, under higher smoothness assumptions, *second-order* methods Nesterov (2021c;a); Kamzolov (2020); Doikov et al. (2024) can surpass the corresponding lower complexity bound for functions with Lipschitz-continuous Hessians. For more comprehensive information on high-order methods, one can refer to the review Kamzolov et al. (2022).

In general, second and higher-order methods are known for their faster convergence compared to first-order methods. However, their computational cost per iteration is significantly higher due to the computation of high-order derivatives. To alleviate this computational burden, it is common to employ approximations of derivatives instead of exact values. While there is a wide range of second-order and tensor methods available for the non-convex case, assuming stochastic or inexact derivatives Cartis et al. (2011a;b); Cartis & Scheinberg (2018); Kohler & Lucchi (2017); Xu et al. (2020); Tripuraneni et al. (2018); Lucchi & Kohler (2023); Bellavia & Gurioli (2022); Bellavia et al. (2022); Doikov et al. (2023), the same cannot be said for the convex case. In the context of convex problems, there have been studies on high-order methods such as second-order methods with inexact Hessian information Ghadimi et al. (2017), tensor methods with inexact and stochastic derivatives Agafonov et al. (2023), and Extra-Newton algorithm with stochastic gradients and Hessians Antonakopoulos et al. (2022). As a possible application of methods with inexact Hessians, we highlight Quasi-Newton(QN) methods. Such methods approximate second-order derivatives using the history of gradient feedback. Quasi-Newton methods are known for their impressive practical performance and local superlinear convergence. However, for the long period of time, the main drawback of such methods was a slow theoretical global convergence, slower than gradient descent. First steps to improve the global convergence of such methods were done in (Scheinberg & Tang, 2016; Ghanbari & Scheinberg, 2018) but the methods could be still slower than gradient descent. The first global Quasi-Newton methods that provably matches the gradient descent were reached by cubic regularization in two consecutive papers Kamzolov et al. (2023); Jiang et al. (2023). It also opened a possibility for accelerated QN Kamzolov et al. (2023) that theoretically matches fast gradient method and first-order lower-bounds. This direction were further explored for different methods in Scieur (2023); Jiang & Mokhtari (2023). Another possible application for high-order methods with inexact or stochastic derivatives is distributed optimization Zhang & Lin (2015); Daneshmand et al. (2021); Agafonov et al. (2021); Dvurechensky et al. (2022).

One of the main challenges of tensor and regularized second-order methods is solving the auxiliary subproblem to compute the iterate update. In both second-order and higher-order cases, it usually requires running a subsolver algorithm. The impact of the accuracy, up to which we solve the auxiliary problem, on the convergence of the algorithm has been studied in several works Grapiglia & Nesterov (2021; 2020); Doikov & Nesterov (2020). One actively developing direction relies on the constructions of CRN with explicit step Polyak (2009; 2017); Mishchenko (2023); Doikov & Nesterov (2023); Doikov et al. (2024); Hanzely et al. (2022).

# B   ON THE INTUITION BEHIND THE ALGORITHM

**Model.** For the second-order case the model $\omega_{x,M}^{\bar{\delta}}(y)$ comprises three key components: an inexact Taylor approximation $\phi_x(y)$; cubic regularization $\frac{M}{6}\|x - y\|^3$ and additional quadratic regularization $\frac{\bar{\delta}}{2}\|x - y\|^2$. The combination of Taylor polynomial and cubic regularization is the standard model for exact second-order methods, as they create a model that is both convex and upper bounds the objective (Nesterov, 2008) (see (Nesterov, 2021b) for high-order optimization). However, inserting

inexactness to the Taylor approximation leads to the necessity of additional regularization (Agafonov et al., 2023).

The first reason to add quadratic regularization is to ensure that the Hessian of the function is majorized by the Hessian of the model:
$$0 \preceq \nabla^2 f(y) \preceq \nabla^2 \phi_x(y) + \delta_2 I + L_2 \|y - x\| \preceq \nabla^2 \phi_x(y) + \bar{\delta}_2 I + M\|y - x\| = \nabla^2 \omega_x^{\bar{\delta}}.$$
Moreover, this regularization is essential for handling stochastic gradients correctly. Note, that we add quadratic regularizer with the constant $\bar{\delta}_t = 2\delta_2 + \frac{\sigma_1}{R}(t+1)^{3/2}$. Here, $\delta_2$ accounts for a Hessian majorization, while $\frac{\sigma_1}{R}(t+1)^{3/2}$ is crucial for achieving optimal convergence in gradient inexactness. From our perspective, this regularization can be viewed as a damping for the size of stochastic Cubic Newton step, as stochastic gradients may lead to undesirable directions.

For further clarification, please refer to Lemma E.3. This lemma serves as a bound on the progress of the step. Take a look at the right-hand side of equation (29). Without proper quadratic regularization, we won't capture the correct term related to Hessian inexactness $\delta_2$. Consequently, the desired convergence term, $\frac{\delta_2 R^2}{T^2}$, cannot be achieved. Moving on to the left-hand side of (29), we encounter the term $\frac{2}{\bar{\delta}}\|g(x) - \nabla f(x)\|$. Here, choosing the appropriate $\bar{\delta}$ is crucial to compensate for stochastic gradient errors and achieve optimal convergence in the gradient inexactness term.

**Estimating sequences.** Estimating sequences are a standard optimization technique to achieve acceleration (Nesterov, 2018)[Section 2.2.1]. As far as our knowledge extends, the application of estimating sequences to second-order methods was first introduced in (Nesterov, 2008). The concept involves adapting acceleration techniques traditionally applied to first-order methods to the realm of second-order methods. In this work, we make slight modifications to the estimating sequences derived from (Nesterov, 2008) to preserve the customary relationships inherent in accelerated methods:
$$\frac{f(x_t)}{A_{t-1}} - err_{low} \leq \psi_t^* \leq \frac{f(x^*)}{A_{t-1}} + c_2\|x^* - x_0\|^2 + c_3\|x^* - x_0\|^3 + err_{up},$$
where $A_t$ and $\alpha_t$ are scaling factors, common for acceleration (Nesterov, 2018; 2008).

For simplicity, let $err_{low} = 0$, $err_{up} = 0$, $c_2 = 0$. That is the case for exact derivatives and subproblem solutions. Then, one can get the convergence, with a specific choice of scaling factors:
$$f(x_t) - f(x^*) \leq A_{t-1}c_3\|x^* - x_0\|^3.$$
In our case, errors and $c_2$ are non zero and stay for gradient, Hessian and subproblem inexactness. By applying estimating sequence technique we get rates (11), (12).

**The choice of parameters.** The cubic regularization parameter $M \geq 2L_2$ represents the standard choice for second-order methods (Nesterov & Polyak, 2006). The quadratic regularization parameter $\bar{\delta}_t = 2\delta_2 + \frac{\sigma_1 + \tau}{R}(t+1)^{3/2}$ consists of $2\delta_2$ for compensating Hessian errors, and $\frac{\sigma_1 + \tau}{R}(t+1)^{3/2}$ for compensating stochastic gradient and subproblem solution errors. The regularization parameters $\bar{\kappa}_2^t, \bar{\kappa}_3^t, \lambda_t$ are utilized for the second step of the method. The $\bar{\kappa}$'s are chosen in (40), (42) to uphold the inequality (34) for acceleration. $\lambda_t = \frac{\sigma}{R}(t+1)^{5/2}$ serves to compensate for stochastic gradient errors in the estimation functions $\psi_t$. The specific choice for $\bar{\delta}$ and $\lambda$ is made in the proof of Theorem E.7 to achieve optimal convergence rate.

## C  ADDITIONAL EXPERIMENTS

After tuning we got the following hyperparameters. For deterministic oracles: $lr = 20$ for GD, $\gamma = 5$ and $\beta_0 = 0.5$ for Extra-Newton, $M = 0.01$ for ASCN. For stochastic oracles: $lr = 20$ for SGD, $\gamma = 5$ and $\beta_0 = 0.05$ for Extra-Newton, $M = 0.01$ and $\sigma = 1e - 7$ for ASCN. For stochastic oracles with different batch sizes: $lr = 20$ for SGD, $\gamma = 5$ and $\beta_0 = 1.0$ for Extra-Newton, $M = 0.01$ and $\sigma = 1e - 7$ for ASCN.

On Figures 3, 4 we present additional experiments with different batch sizes for gradients and Hessians, and on Figures 5, 6 we present Figure 2 with increased size. Moving forward to Figures 7 and 8, a comparison in running time is illustrated for two distinct setups: the gradient batch size is set at 10000, and Hessian batch sizes are configured to be 150 and 450. Specifically, one iteration of SGD consumes 0.16 seconds, while the execution times for EN and ASCN are approximately 0.33 seconds for both scenarios.

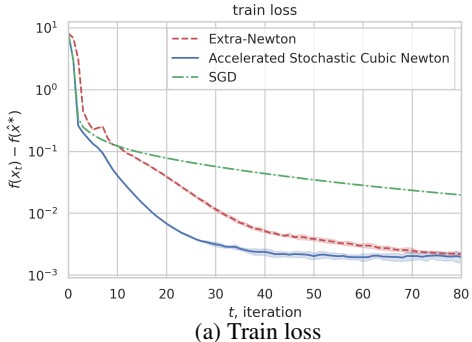 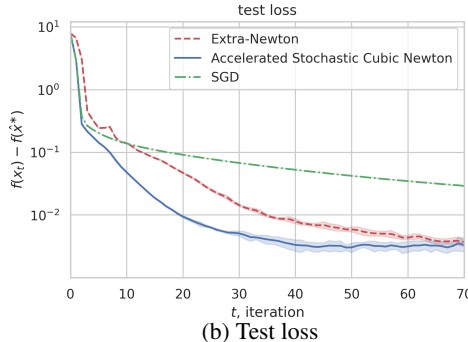

(a) Train loss  (b) Test loss

Figure 3: Logistic regression on `a9a`. Gradient batch size is 10000, Hessian batch size is 450

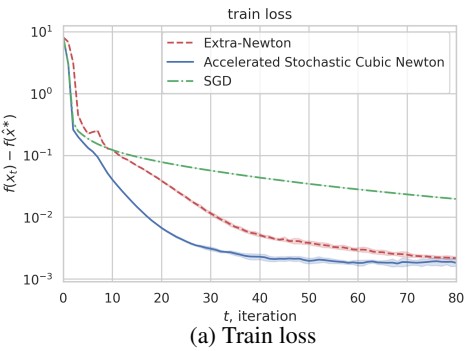 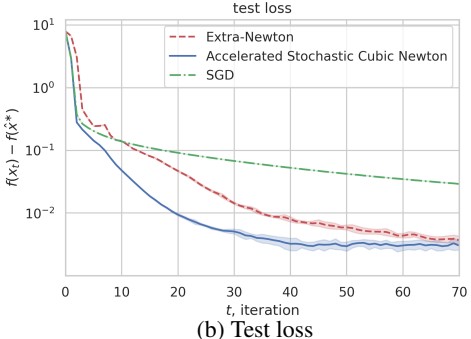

(a) Train loss  (b) Test loss

Figure 4: Logistic regression on `a9a`. Gradient batch size is 10000, Hessian batch size is 900

## D  PROOF OF LEMMAS 2.1 AND 5.2

**Lemma 5.2.** *Let Assumption 5.1 hold. Then, for any* $x, y \in \mathbb{R}^d$, *we have*

$$|f(y) - \phi_{x,p}(y)| \leq \sum_{i=1}^{p} \frac{1}{i!} \|(G_i(x) - \nabla^i f(x))[y - x]^{i-1}\| \|y - x\| + \frac{L_p}{(p+1)!} \|y - x\|^{p+1},$$
$$\|\nabla f(y) - \nabla \phi_{x,p}(y)\| \leq \sum_{i=1}^{p} \frac{1}{(i-1)!} \|(G_i(x) - \nabla^i f(x))[y - x]^{i-1}\| + \frac{L_p}{p!} \|y - x\|^p,$$

The proof of that lemma is the same as proofs of Lemmas 1, 2 from Agafonov et al. (2023). Lemma 2.1 is a special case of the Lemma 5.2 for $p = 2$.

## E  PROOF OF THEOREM 3.2

The full proof is organized as follows:

- Lemma E.1 provides an upper bound for the estimating sequence $\psi_t(x)$;
- Lemmas E.2, E.6 present the efficiency of Inexact Cubic Newton step $x_{t+1} = S_{M,\delta_t}(v_t)$.
- Lemma E.6 provides a lower bound on $\psi_t(x)$ based on results of technical Lemmas E.5-E.4;
- Everything is combined together in Theorem E.7 in order to prove convergence and obtain convergence rate.

The following lemma shows that the sequence of functions $\bar{\psi}_t(x)$ can be upper bounded by the properly regularized objective function.

**Lemma E.1.** *For convex function* $f(x)$ *and* $\psi_t(x)$, *we have*

$$\psi_t(x^*) \leq \frac{f(x^*)}{A_{t-1}} + \frac{\bar{\kappa}_2^t + \lambda_t}{2} \|x^* - x_0\|^2 + \frac{\bar{\kappa}_3^t}{6} \|x^* - x_0\|^3 + err_t^{up}, \tag{21}$$

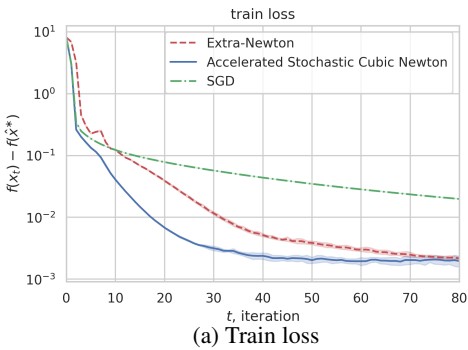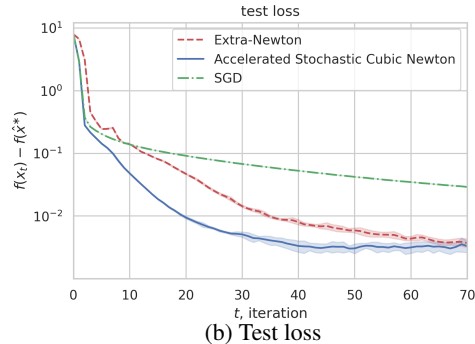

(a) Train loss

(b) Test loss

Figure 5: Logistic regression on `a9a`. Gradient and Hessian batch sizes are 1500

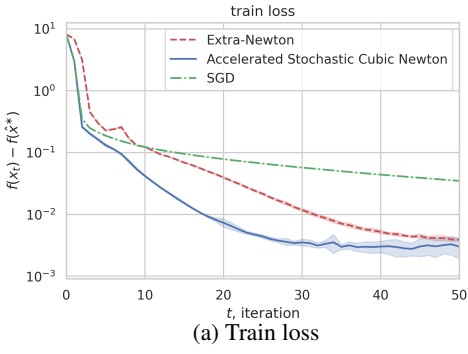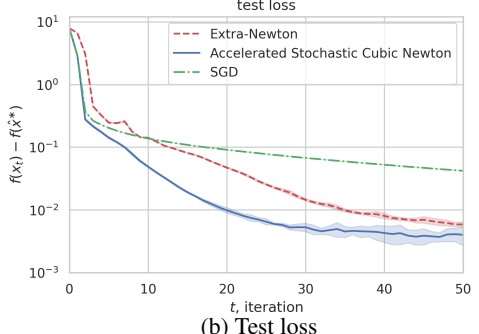

(a) Train loss

(b) Test loss

Figure 6: Logistic regression on `a9a`. Gradient batch size is 10000, Hessian batch size is 150

*where*

$$err_t^{up} = \sum_{j=0}^{t-1} \frac{\alpha_j}{A_j} \langle g(x_{j+1}) - \nabla f(x_{j+1}), x^* - x_{j+1} \rangle \qquad (22)$$

*Proof.* For $t = 0$, let us define $A_{-1}$ such that $\frac{1}{A_{-1}} = 0$ then $\frac{f(x^*)}{A_{-1}} = 0$ and

$$\psi_0(x^*) \le \frac{\bar{\kappa}_2^0 + \lambda_0}{2} \|x^* - x_0\|^2 + \frac{\bar{\kappa}_3^0}{3} \|x^* - x_0\|^3$$

From

$$\psi_{t+1}(x) := \psi_t(x) + \frac{\lambda_{t+1} - \lambda_t}{2} \|x - x_0\|^2 + \sum_{i=2}^{3} \frac{\bar{\kappa}_i^{t+1} - \bar{\kappa}_i^t}{i} \|x - x_0\|^i + \frac{\alpha_t}{A_t} l(x, x_{t+1}), \qquad (23)$$

we have

$$\psi_t(x^*) = \frac{\bar{\kappa}_2^t + \lambda_t}{2} \|x^* - x_0\|^2 + \frac{\bar{\kappa}_3^t}{3} \|x^* - x_0\|^3 + \sum_{j=0}^{t-1} \frac{\alpha_j}{A_j} l(x^*, x_{j+1}). \qquad (24)$$

From (7), we have that, for all $j \ge 1$, $A_j = A_{j-1}(1 - \alpha_j)$, which leads to $\frac{\alpha_j}{A_j} = \frac{1}{A_j} - \frac{1}{A_{j-1}}$. Hence, we have $\sum_{j=0}^{t-1} \frac{\alpha_j}{A_j} = \frac{1}{A_{t-1}} - \frac{1}{A_{-1}} = \frac{1}{A_{t-1}}$ and, using the convexity of the objective $f$, we get

$$\sum_{j=0}^{t-1} \frac{\alpha_j}{A_j} l(x^*, x_{j+1}) \le \sum_{j=0}^{t-1} \frac{\alpha_j}{A_j} \bar{l}(x^*, x_{j+1}) + \sum_{j=0}^{t-1} \frac{\alpha_j}{A_j} \langle g(x_{j+1}) - \nabla f(x_{j+1}), x^* - x_{j+1} \rangle \qquad (25)$$

$$\le f(x^*) \sum_{j=0}^{t-1} \frac{\alpha_j}{A_j} + err_t^{up} = \frac{f(x^*)}{A_{t-1}} + err_t^{up}, \qquad (26)$$

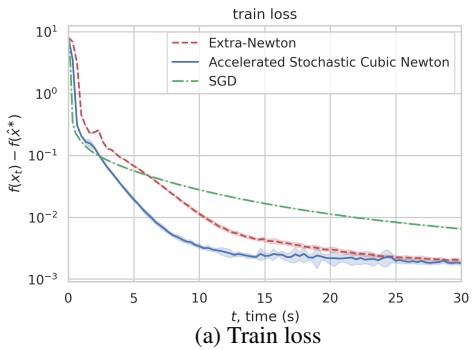 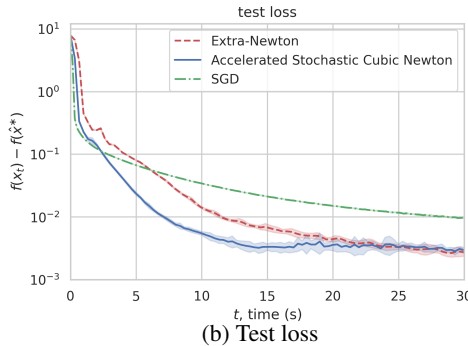

Figure 7: Logistic regression on `a9a`. Gradient batch size is 10000, Hessian batch size is 150

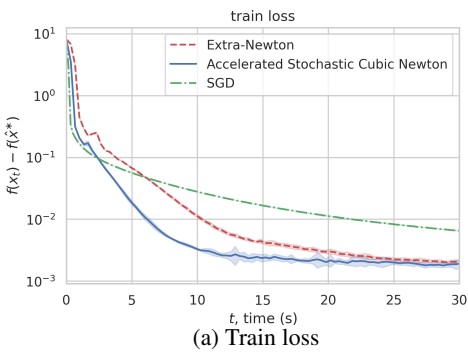 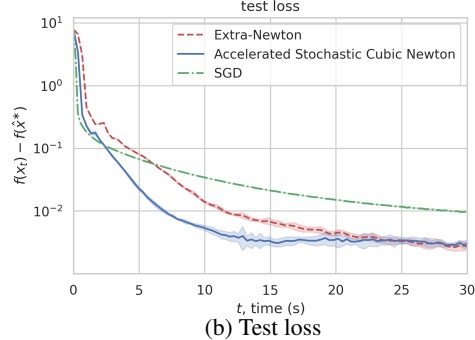

Figure 8: Logistic regression on `a9a`. Gradient batch size is 10000, Hessian batch size is 450

where $\bar{l}(x, y) = f(y) + \langle \nabla f(y), x - y \rangle$. Finally, combining all the inequalities from above, we obtain

$$\psi_t(x^*) \overset{(24),(26)}{\leq} \frac{f(x^*)}{A_{t-1}} + \frac{\bar{\kappa}_2^t + \lambda_t}{2} \|x^* - x_0\|^2 + \frac{\bar{\kappa}_3^t}{6} \|x^* - x_0\|^3 + err_t^{up}. \tag{27}$$

$\square$

**Lemma E.2.** *For the function $f(x)$ with $L_2$-Lipschitz-continuous Hessian and $H(x_t)$ is $\delta_2^t$-inexact Hessian for $v_t, x_{t+1} \in \mathbb{R}^d$ we have*

$$\|\nabla\phi_{v_t}(x_{t+1}) - \nabla f(x_{t+1})\| \leq \delta_2^t \|x_{t+1} - v_t\| + \frac{L_2}{2} \|x_{t+1} - v_t\|^2 + \|g(v_t) - \nabla f(v_t)\|, \tag{28}$$

*where we denote $\delta_2^t = \delta_t^{v_t, x_{t+1}}$ to simplify the notation.*

*Proof.*
$$
\begin{aligned}
\|\nabla\phi_{v_t}(x_{t+1}) - \nabla f(x_{t+1})\| &= \|\nabla\phi_{v_t}(x_{t+1}) - \Phi_{v_t}(x_{t+1}) + \Phi_{v_t}(x_{t+1}) - \nabla f(x_{t+1})\| \\
&\leq \|\nabla\phi_{v_t}(x_{t+1}) - \Phi_{v_t}(x_{t+1})\| + \|\Phi_{v_t}(x_{t+1}) - \nabla f(x_{t+1})\| \\
&= \|(\nabla^2 f(v_t) - B_{v_t})(x_{t+1} - v_t)\| + \|\Phi_{v_t}(x_{t+1}) - \nabla f(x_{t+1})\| \\
&\quad + \|g(v_t) - \nabla f(v_t)\| \\
&\leq \delta_2^t \|x_{t+1} - v_t\| + \frac{L_2}{2} \|x_{t+1} - v_t\|^2 + \|g(v_t) - \nabla f(v_t)\|
\end{aligned}
$$
$\square$

The next Lemma characterizes the progress of the inexact cubic step $S_{M, \delta_t}(v_t)$ in Algorithm 1.

**Lemma E.3.** *Let $\{x_t, v_t\}_{t \geq 1}$ be generated by Algorithm 1. Then, for any $\bar{\delta}_t \geq 4\delta_2^t$, $M \geq 4L_2$ and $x_{t+1} = S_{M, \bar{\delta}_t}(v_t)$, the following holds*

$$\frac{\tau^2}{\bar{\delta}_t} + \frac{2}{\bar{\delta}_t}\|g(v_t) - \nabla f(v_t)\|^2 + \langle \nabla f(x_{t+1}), v_t - x_{t+1}\rangle$$

$$\geq \min\left\{\|\nabla f(x_{t+1})\|^2\left(\frac{1}{4\delta_t}\right), \ \|\nabla f(x_{t+1})\|^{\frac{3}{2}}\left(\frac{1}{3M}\right)^{\frac{1}{2}}\right\}. \tag{29}$$

*Proof.* For simplicity, we denote $r_{t+1} = \|x_{t+1} - v_t\|$ and

$$\zeta_{t+1} = \bar{\delta}_t + \frac{M}{2}\|x_{t+1} - v_t\|. \tag{30}$$

By Definition 3.1 for $x_{t+1} = S^{M, \bar{\delta}_t, \tau}(v_t)$

$$\tau \geq \|\nabla\phi_{v_t}(x_{t+1}) + \bar{\delta}_t(x_{t+1} - v_t) + \frac{M}{2}\|x_{t+1} - v_t\|(x_{t+1} - v_t)\|$$

$$\overset{(30)}{=} \|\nabla\phi_{v_t}(x_{t+1}) + \zeta_{t+1}(x_{t+1} - v_t)\|. \tag{31}$$

We start with getting an upper bound for $\|\nabla\phi_{v_t}(x_{t+1}) - \nabla f(x_{t+1})\|$.

$$\|\nabla\phi_{v_t}(x_{t+1}) - \nabla f(x_{t+1})\| \overset{(28)}{\leq} \delta_2^t r_{t+1} + \frac{L_2}{2}r_{t+1}^2 + \|g(v_t) - \nabla f(v_t)\|. \tag{32}$$

From inexact solution of the subproblem we get

$$\|\nabla f(x_{k+1}) + \zeta_{t+1}(x_{t+1} - v_t)\|^2$$

$$\leq \|\nabla\phi_{v_t}(x_{t+1}) - \nabla f(x_{t+1}) - \nabla\phi_{v_t}(x_{t+1}) - \zeta_{t+1}(x_{t+1} - v_t)\|^2$$

$$\leq 2\|\nabla\phi_{v_t}(x_{t+1}) - \nabla f(x_{t+1})\|^2 + 2\|\nabla\phi_{v_t}(x_{t+1}) + \zeta_{t+1}(x_{t+1} - v_t)\|^2 \tag{33}$$

$$\overset{\text{Def. 3.1}}{\leq} 2\|\nabla\phi_{v_t}(x_{t+1}) - \nabla f(x_{t+1})\|^2 + 2\tau^2$$

Next, from the previous inequality and (32), we get

$$4\|g(v_t) - \nabla f(v_t)\|^2 + 4(\delta_2^t + \frac{L_2}{2}r_{t+1})^2 r_{t+1}^2 \overset{(32)}{\geq} 2\|\nabla\phi_{v_t,p}(x_{t+1}) - \nabla f(x_{t+1})\|^2$$

$$\overset{(33)}{\geq} \|\nabla f(x_{t+1}) + \zeta_{t+1}(x_{t+1} - v_t)\|^2 - 2\tau^2$$

$$= 2\langle\nabla f(x_{t+1}), x_{t+1} - v_t\rangle\zeta_{t+1} + \|\nabla f(x_{t+1})\|^2 + \zeta_{t+1}^2\|x_{t+1} - v_t\|^2 - 2\tau^2.$$

Hence,

$$\frac{\tau^2}{\zeta_{t+1}} + \frac{2}{\zeta_{t+1}}\|g(v_t) - \nabla f(v_t)\|^2 + \langle\nabla f(x_{t+1}), v_t - x_{t+1}\rangle \geq \frac{1}{2\zeta_{t+1}}\|\nabla f(x_{t+1})\|^2 + \frac{1}{2\zeta_{t+1}}\left(\zeta_{t+1}^2 - 4(\delta_2^t + \frac{L_2}{2}r_{t+1})^2\right)r_{t+1}^2,$$

and finally by using defenetion of $\zeta_{t+1}$, we get

$$\frac{\tau^2}{\bar{\delta}_t} + \frac{2}{\bar{\delta}_t}\|g(v_t) - \nabla f(v_t)\|^2 + \langle\nabla f(x_{t+1}), v_t - x_{t+1}\rangle \geq \frac{1}{2\zeta_{t+1}}\|\nabla f(x_{t+1})\|^2 + \frac{1}{2\zeta_{t+1}}\left(\zeta_{t+1}^2 - 4(\delta_2^t + \frac{L_2}{2}r_{t+1})^2\right)r_{t+1}^2$$

Next, we consider 2 cases depending on which term dominates in the $\zeta_{t+1}$.

- If $\bar{\delta}_t \geq \frac{M}{2}\|x_{t+1} - v_t\|$, then we get the following bound

$$\frac{\tau^2}{\bar{\delta}_t} + \frac{2}{\bar{\delta}_t}\|g(v_t) - \nabla f(v_t)\|^2 + \langle\nabla f(x_{t+1}), v_t - x_{t+1}\rangle \geq \frac{1}{2\zeta_{t+1}}\|\nabla f(x_{t+1})\|^2 + \frac{1}{2\zeta_{t+1}}\left(\zeta_{t+1}^2 - 4(\delta_2^t + \frac{L_2}{2}r_{t+1})^2\right)r_{t+1}^2$$

$$\geq \frac{1}{4\delta_t}\|\nabla f(x_{t+1})\|^2$$

- If $\bar{\delta}_t < \frac{M}{2}\|x_{t+1} - v_t\|$, then similarly to previous case, we get

$$\frac{\tau^2}{\bar{\delta}_t} + \frac{2}{\bar{\delta}_t}\|g(v_t) - \nabla f(v_t)\|^2 + \langle\nabla f(x_{t+1}), v_t - x_{t+1}\rangle$$

$$\geq \frac{\|\nabla f(x_{t+1})\|^2}{2\zeta_{t+1}} + \left(\left(\bar{\delta}_t + \frac{M}{2}r_{t+1}\right)^2 - 4\left(\delta_2^t + \frac{L_2}{2}r_{t+1}\right)^2\right)\frac{r_{t+1}^2}{2\zeta_{t+1}}$$

$$= \frac{\|\nabla f(x_{t+1})\|^2}{2\zeta_{t+1}} + \left(\left(\bar{\delta}_t - 2\delta_2^t + \frac{M-2L_2}{2}r_{t+1}\right)\left(\bar{\delta}_t + 2\delta_2^t + \frac{2L_2+M}{2}r_{t+1}\right)\right)\frac{r_{t+1}^2}{2\zeta_{t+1}}$$

$$\geq \frac{\|\nabla f(x_{t+1})\|^2}{2Mr_{t+1}} + \frac{M^2 - 4L_2^2}{4} \frac{r_{t+1}^3}{2M} \geq \frac{\|\nabla f(x_{t+1})\|^2}{2Mr_{t+1}} + \frac{2M}{32} r_{t+1}^3 \geq \left(\frac{1}{3M}\right)^{\frac{1}{2}} \|\nabla f(x_{t+1})\|^{\frac{3}{2}},$$

where for the last inequality, we use $\frac{\alpha}{r} + \frac{\beta r^3}{3} \geq \frac{4}{3}\beta^{1/4}\alpha^{3/4}$.

$\square$

We use the following lemma (Agafonov et al., 2023, Lemma 7).

**Lemma E.4.** *Let $h(x)$ be a convex function, $x_0 \in \mathbb{R}^n$, $\theta_i \geq 0$ for $i = 2, \ldots, p + 1$ and*

$$\bar{x} = \arg\min_{x \in \mathbb{R}^n} \{\bar{h}(x) = h(x) + \sum_{i=2}^{p+1} \theta_i d_i(x - x_0)\},$$

*where $d_i(x) = \frac{1}{i}\|x\|^i$ is a power-prox function. Then, for all $x \in \mathbb{R}^n$,*

$$\bar{h}(x) \geq \bar{h}(\bar{x}) + \sum_{i=2}^{p+1} \left(\frac{1}{2}\right)^{i-2} \theta_i d_i(x - \bar{x}).$$

We will also use the next technical lemma Nesterov (2008); Ghadimi et al. (2017) on Fenchel conjugate for the $p$-th power of the norm.

**Lemma E.5.** *Let $g(z) = \frac{\theta}{p}\|z\|^p$ for $p \geq 2$ and $g^*$ be its conjugate function i.e., $g^*(v) = \sup_z\{\langle v, z\rangle - g(z)\}$. Then, we have*

$$g^*(v) = \frac{p-1}{p}\left(\frac{\|v\|^p}{\theta}\right)^{\frac{1}{p-1}}$$

*Moreover, for any $v, z \in \mathbb{R}^n$, we have $g(z) + g^*(v) - \langle z, v\rangle \geq 0$.*

Finally, the last step is the next Lemma which prove that $\frac{f(x_t)}{A_{t-1}} \leq \min_x \bar{\psi}_t(x) = \bar{\psi}_t^*$.

**Lemma E.6.** *Let $\{x_t, y_t\}_{t \geq 1}$ be generated by Algorithm 1. Then*

$$\psi_t^* = \min_x \psi_t(x) \geq \frac{f(x_t)}{A_{t-1}} - err_t^v - err_t^x - err_t^\tau, \tag{34}$$

*where*

$$err_t^v = \sum_{j=0}^{t-1} \frac{2}{A_j\bar{\delta}_j}\|g(v_j) - \nabla f(v_j)\|^2, \tag{35}$$

$$err_t^\tau = \sum_{j=0}^{t-1} \frac{\tau^2}{A_j\bar{\delta}_j}, \tag{36}$$

*and*

$$err_t^x = \sum_{j=0}^{t-1} \frac{\alpha_j^2}{2A_j^2\lambda_j}\|g(x_{j+1}) - \nabla f(x_{j+1})\|^2 + \sum_{j=0}^{t-1} \frac{\alpha_j}{A_j}\langle g(x_{j+1}) - \nabla f(x_{j+1}), y_j - x_{j+1}\rangle. \tag{37}$$

*Proof.* We prove Lemma by induction. Let us start with $t = 0$, we define $A_{-1}$ such that $\frac{1}{A_{-1}} = 0$. Then $\frac{f(x_0)}{A_{-1}} = 0$ and $\psi_0^* = 0$, hence, $\psi_0^* \geq \frac{f(x_0)}{A_{-1}}$. Let us assume that (34) is true for $t$ and show that (34) is true for $t + 1$. By definition,

$$\psi_t(x) = \frac{\lambda_t + \bar{\kappa}_2^t}{2}\|x - x_0\|^2 + \frac{\bar{\kappa}_3^t}{3}\|x - x_0\|^3 + \sum_{j=0}^{t-1} \frac{\alpha_j}{A_j} l(x, x_{j+1}).$$

Next, we apply Lemma E.4 with the following choice of parameters: $h(x) = \sum_{j=0}^{t-1} \frac{\alpha_j}{A_j} l(x, x_{j+1})$, $\theta_2 = \lambda_t + \bar{\kappa}_2^t$, and $\theta_3 = \bar{\kappa}_3^t$.

By (23), $y_t = \underset{x \in \mathbb{R}^d}{\operatorname{argmin}} h(x)$, and we have

$$\psi_t(y_{t+1}) \geq \psi_t^* + \frac{\bar{\kappa}_2^t + \lambda_t}{2} \|x - y_t\|^2 + \frac{\bar{\kappa}_3^t}{6} \|x - y_t\|^3$$

$$\overset{(34)}{\geq} \frac{f(x_t)}{A_{t-1}} + \frac{\bar{\kappa}_2^t + \lambda_t}{2} \|x - y_t\|^2 + \frac{\bar{\kappa}_3^t}{6} \|x - y_t\|^3 - err_t^v - err_t^x - err_t^\tau,$$

where the last inequality follows from the assumption of the lemma.

By the definition of $\psi_{t+1}(x)$, the above inequality, and convexity of $f$, we obtain

$$\psi_{t+1}(y_{t+1}) = \psi_t(y_{t+1}) + \frac{\lambda_{t+1} - \lambda_t}{2} \|y_{t+1} - x_0\|^2 + \frac{\bar{\kappa}_2^{t+1} - \bar{\kappa}_2^t}{2} \|y_{t+1} - x_0\|^2$$

$$+ \frac{\bar{\kappa}_3^{t+1} - \bar{\kappa}_3^t}{3} \|y_{t+1} - x_0\|^3 + \frac{\alpha_t}{A_t} l(y_{t+1}, x_{t+1})$$

$$\geq \frac{f(x_t)}{A_{t-1}} + \frac{\bar{\kappa}_2^t + \lambda_t}{2} \|y_{t+1} - y_t\|^2 + \frac{\bar{\kappa}_3^t}{6} \|y_{t+1} - y_t\|^3 + \frac{\alpha_t}{A_t} l(y_{t+1}, x_{t+1}) - err_t^v - err_t^x - err_t^\tau$$

$$= \frac{f(x_t)}{A_{t-1}} + \frac{\bar{\kappa}_2^t + \lambda_t}{2} \|y_{t+1} - y_t\|^2 + \frac{\bar{\kappa}_3^t}{6} \|y_{t+1} - y_t\|^3 - err_t^v - err_t^x - err_t^\tau$$

$$\frac{\alpha_t}{A_t} (f(x_{t+1}) + \langle \nabla f(x_{t+1}), y_{t+1} - x_{t+1} \rangle + \langle g(x_{t+1}) - \nabla f(x_{t+1}), y_{t+1} - y_t \rangle + \langle g(x_{t+1}) - \nabla f(x_{t+1}), y_t - x_{t+1} \rangle)$$

$$\geq \frac{f(x_t)}{A_{t-1}} + \frac{\bar{\kappa}_2^t}{2} \|y_{t+1} - y_t\|^2 + \frac{\bar{\kappa}_3^t}{6} \|y_{t+1} - y_t\|^3 - err_t^v - err_t^x - err_t^\tau$$

$$+ \frac{\alpha_t}{A_t} \left( f(x_{t+1}) + \langle \nabla f(x_{t+1}), y_{t+1} - x_{t+1} \rangle \right) - \frac{\alpha_t^2}{2A_t^2 \lambda_t} \|g(x_{t+1}) - \nabla f(x_{t+1})\|^2 + \frac{\alpha_t}{A_t} \langle g(x_{t+1}) - \nabla f(x_{t+1}), y_t - x_{t+1} \rangle$$

$$= \frac{f(x_t)}{A_{t-1}} + \frac{\bar{\kappa}_2^t}{2} \|y_{t+1} - y_t\|^2 + \frac{\bar{\kappa}_3^t}{6} \|y_{t+1} - y_t\|^3 - err_t^v - err_{t+1}^x - err_t^\tau + \frac{\alpha_t}{A_t} \left( f(x_{t+1}) + \langle \nabla f(x_{t+1}), y_{t+1} - x_{t+1} \rangle \right)$$

$$\geq \frac{\bar{\kappa}_2^t}{2} \|y_{t+1} - y_t\|^2 + \frac{\bar{\kappa}_3^t}{6} \|y_{t+1} - y_t\|^3 - err_t^v - err_{t+1}^x - err_t^\tau$$

$$+ \frac{1}{A_{t-1}} (f(x_{t+1}) + \langle \nabla f(x_{t+1}), x_t - x_{t+1} \rangle) + \frac{\alpha_t}{A_t} \left( f(x_{t+1}) + \langle \nabla f(x_{t+1}), y_{t+1} - x_{t+1} \rangle \right).$$

Next, we consider the sum of two linear models from the last inequality:

$$\frac{1}{A_{t-1}} (f(x_{t+1}) + \langle \nabla f(x_{t+1}), x_t - x_{t+1} \rangle) + \frac{\alpha_t}{A_t} (f(x_{t+1}) + \langle \nabla f(x_{t+1}), y_{t+1} - x_{t+1} \rangle)$$

$$\overset{(7)}{=} \frac{1 - \alpha_t}{A_t} f(x_{t+1}) + \frac{1 - \alpha_t}{A_t} \langle \nabla f(x_{t+1}), x_t - x_{t+1} \rangle + \frac{\alpha_t}{A_t} f(x_{t+1}) + \frac{\alpha_t}{A_t} \langle \nabla f(x_{t+1}), y_{t+1} - x_{t+1} \rangle$$

$$= \frac{f(x_{t+1})}{A_t} + \frac{1 - \alpha_t}{A_t} \langle \nabla f(x_{t+1}), \frac{v_t - \alpha_t y_t}{1 - \alpha_t} - x_{t+1} \rangle + \frac{\alpha_t}{A_t} \langle \nabla f(x_{t+1}), y_{t+1} - x_{t+1} \rangle$$

$$= \frac{f(x_{t+1})}{A_t} + \frac{1}{A_t} \langle \nabla f(x_{t+1}), v_t - x_{t+1} \rangle + \frac{\alpha_t}{A_t} \langle \nabla f(x_{t+1}), y_{t+1} - y_t \rangle.$$

As a result, by (23), we get

$$\psi_{t+1}^* = \psi_{t+1}(y_{t+1}) \geq \frac{f(x_{t+1})}{A_t} + \frac{1}{A_t} \langle \nabla f(x_{t+1}), v_t - x_{t+1} \rangle + \frac{\bar{\kappa}_2^t}{2} \|y_{t+1} - y_t\|^2$$

$$+ \frac{\bar{\kappa}_3^t}{6} \|y_{t+1} - y_t\|^3 + \frac{\alpha_t}{A_t} \langle \nabla f(x_{t+1}), y_{t+1} - y_t \rangle - err_t^v - err_{t+1}^x - err_t^\tau. \tag{38}$$

To complete the induction step, we show, that the sum of all terms in the RHS except $\frac{f(x_{t+1})}{A_t}$ is non-negative (except err).

Lemma E.3 provides the lower bound for $\langle \nabla f(x_{t+1}), v_t - x_{t+1} \rangle$. Let us consider the case when the minimum in the RHS of (29) is attained at the first term. By Lemma E.5 with the following choice of the parameters

$$z = y_t - y_{t+1}, \quad v = \frac{\alpha_t}{A_t} \nabla f(x_{t+1}), \quad \theta = \bar{\kappa}_i^t,$$

we have

$$\frac{\bar{\kappa}_2^t}{2}\|y_t - y_{t+1}\|^2 + \frac{\alpha_t}{A_t}\langle \nabla f(x_{t+1}), y_{t+1} - y_t\rangle \geq -\frac{1}{2}\left(\frac{\|\frac{\alpha_t}{A_t}\nabla f(x_{t+1})\|^2}{\bar{\kappa}_2^t}\right). \tag{39}$$

Hence,

$$\frac{1}{A_t}\langle \nabla f(x_{t+1}), v_t - x_{t+1}\rangle + \frac{\bar{\kappa}_2^t}{2}\|y_{t+1} - y_t\|^2 + \frac{\alpha_t}{A_t}\langle \nabla f(x_{t+1}), y_{t+1} - y_t\rangle$$

$$\overset{(39)}{\geq} \frac{1}{A_t}\langle \nabla f(x_{t+1}), v_t - x_{t+1}\rangle - \frac{\|\frac{\alpha_t}{A_t}\nabla f(x_{t+1})\|^i}{2\bar{\kappa}_2^t}$$

$$\overset{(29)}{\geq} \frac{1}{A_t}\|\nabla f(x_{t+1})\|^2\left(\frac{1}{4\bar{\delta}_t}\right) - \frac{\|\frac{\alpha_t}{A_t}\nabla f(x_{t+1})\|^2}{2\bar{\kappa}_2^{t+1}} - \frac{2}{A_t\bar{\delta}_t}\|g(v_t) - \nabla f(v_t)\|^2 - \frac{\tau^2}{A_t\bar{\delta}_t}$$

$$\geq -\frac{2}{A_t\bar{\delta}_t}\|g(v_t) - \nabla f(v_t)\|^2 - \frac{\tau^2}{A_t\bar{\delta}_t},$$

where the last inequality holds by our choice of the parameters

$$\bar{\kappa}_2^{t+1} \geq \frac{2\bar{\delta}_t\alpha_t^2}{A_t}. \tag{40}$$

Next, we consider the case when the minimum in the RHS of (29) is achieved on the second term. Again, by Lemma E.5 with the same choice of $z, v$ and with $\theta = \frac{\bar{\kappa}_3^t}{2}$, we have

$$\frac{\bar{\kappa}_3^t}{6}\|y_t - y_{t+1}\|^3 + \frac{\alpha_t}{A_t}\langle \nabla f(x_{t+1}), y_{t+1} - y_t\rangle \geq -\frac{2}{3}\left(\frac{2\|\frac{\alpha_t}{A_t}\nabla f(x_{t+1})\|^3}{\bar{\kappa}_3^t}\right)^{\frac{1}{2}}. \tag{41}$$

Hence, we get

$$\frac{1}{A_t}\langle \nabla f(x_{t+1}), v_t - x_{t+1}\rangle + \frac{\bar{\kappa}_3^t}{6}\|y_{t+1} - y_t\|^3 + \frac{\alpha_t}{A_t}\langle \nabla f(x_{t+1}), y_{t+1} - y_t\rangle$$

$$\overset{(41)}{\geq} \frac{1}{A_t}\langle \nabla f(x_{t+1}), v_t - x_{t+1}\rangle - \frac{2}{3}\left(\frac{2\|\frac{\alpha_t}{A_t}\nabla f(x_{t+1})\|^3}{\bar{\kappa}_3^t}\right)^{\frac{1}{2}}$$

$$\overset{(29)}{\geq} \frac{1}{A_t}\|\nabla f(x_{t+1})\|^{\frac{3}{2}}\left(\frac{1}{3M}\right)^{\frac{1}{2}} - \frac{2}{3}\left(\frac{2\|\frac{\alpha_t}{A_t}\nabla f(x_{t+1})\|^3}{\bar{\kappa}_3^t}\right)^{\frac{1}{2}} - \frac{2}{A_t\bar{\delta}_t}\|g(v_t) - \nabla f(v_t)\|^2 - \frac{\tau^2}{A_t\bar{\delta}_t}$$

$$\geq -\frac{2}{A_t\bar{\delta}_t}\|g(v_t) - \nabla f(v_t)\|^2 - \frac{\tau^2}{A_t\bar{\delta}_t},$$

where the last inequality holds by our choice of $\bar{\kappa}_3^t$:

$$\bar{\kappa}_3^t \geq \frac{8M}{3}\frac{\alpha_t^3}{A_t}. \tag{42}$$

As a result, we unite both cases and get

$$\psi_{t+1}^* \geq \frac{f(x_{t+1})}{A_t} + \frac{1}{A_t}\langle \nabla f(x_{t+1}), v_t - x_{t+1}\rangle + \frac{\bar{\kappa}_2^t}{2}\|y_{t+1} - y_t\|^2$$

$$+\frac{\bar{\kappa}_3^t}{6}\|y_{t+1} - y_t\|^3 + \frac{\alpha_t}{A_t}\langle \nabla f(x_{t+1}), y_{t+1} - y_t\rangle - err_t^v - err_{t+1}^x - err_t^\tau$$

$$\geq \frac{f(x_{t+1})}{A_t} - \frac{2}{A_t\bar{\delta}_t}\|g(v_t) - \nabla f(v_t)\|^2 - \frac{\tau^2}{A_t\bar{\delta}_t} - err_t^v - err_{t+1}^x$$

$$= \frac{f(x_{t+1})}{A_t} - err_{t+1}^v - err_{t+1}^x - err_{t+1}^\tau$$

To sum up, by our choice of the parameters $\bar{\kappa}_i^t$, $i = 2, 3$, we prove the induction step.

$$\square$$

Finally, we are in a position to prove the convergence rate theorem The proof uses the following technical assumption. Let $R$ be such that

$$\|x_0 - x^*\| \leq R. \tag{43}$$

**Theorem E.7.** *Let Assumption 1.1 hold and $M \geq 4L_2$. Let Assumption 2.3 hold. After $T \geq 1$ with parameters defined in (10) and $\sigma_2 = \delta_2 = \max_{t=1,\ldots,T} \delta_t^{v_{t-1}, x_t}$ we get the following bound for the objective residual*

$$\mathbb{E}\left[f(x_T) - f(x^*)\right] \leq \frac{10\tau R}{\sqrt{T+2}} + \frac{19\sigma_1 R}{\sqrt{T+1}} + \frac{18\delta_2 R^2}{(T+3)^2} + \frac{20L_2 R^3}{(T+1)^3}.$$

*Proof.* First of all, let us bound $A_T$.

$$\alpha_t = \frac{3}{t+3}, \ t \geq 1. \tag{44}$$

Then, we have

$$A_T = \prod_{t=1}^{T}(1 - \alpha_t) = \prod_{t=1}^{T} \frac{t}{t+3} = \frac{T!3!}{(T+3)!} = \frac{6}{(T+1)(T+2)(T+3)}. \tag{45}$$

And from Agafonov et al. (2023) we get

$$\sum_{t=0}^{T} \frac{A_T \alpha_t^i}{A_t} \leq \frac{3^i}{(T+3)^{i-1}} \tag{46}$$

From Lemmas E.6 and E.1, we obtain that, for all $t \geq 1$,

$$\frac{f(x_{t+1})}{A_t} - err_{t+1}^v - err_{t+1}^x - err_{t+1}^\tau \overset{(34)}{\leq} \psi_{t+1}^* \leq \psi_{t+1}(x^*)$$

$$\overset{(21)}{\leq} \frac{f(x^*)}{A_t} + \frac{\bar{\kappa}_2^t + \lambda_t}{2}\|x^* - x_0\|^2 + \frac{\bar{\kappa}_3^t}{6}\|x^* - x_0\|^3 + err_{t+1}^{up}.$$

Next, we apply expectation

$$\mathbb{E}\left[f(x_{T+1}) - f(x^*)\right] \leq A_T \mathbb{E}\left[\frac{\bar{\kappa}_2^T + \lambda_T}{2}R^2 + \frac{\bar{\kappa}_3^T}{6}R^3 + err_{T+1}^{up} + err_{T+1}^v + err_{T+1}^x + err_{T+1}^\tau\right]. \tag{47}$$

Let us choose

$$\bar{\delta}_t = \delta_2 + \frac{\tau + \sigma}{R}(t+3)^{3/2}, \tag{48}$$

$$\lambda_t = \frac{\sigma}{R}(t+3)^{5/2}. \tag{49}$$

Then, we bound terms in (47) step by step. We start from deterministic terms.

$$A_T \mathbb{E}\left[\frac{\bar{\kappa}_2^T + \lambda_T}{2}R^2 + \frac{\bar{\kappa}_3^T}{6}R^3\right] \overset{(10)}{=} \frac{18\bar{\delta}_T R^2}{(T+3)^2} + \frac{6\lambda_T R^2}{(T+3)^3} + \frac{72MR^3}{(T+3)^3}$$

$$\overset{(48),(49)}{=} \frac{18\tau R}{(T+3)^{1/2}} + \frac{18\sigma R}{(T+3)^{1/2}} + \frac{18\delta_2 R^2}{(T+3)^2} + \frac{6\sigma R}{(T+3)^{1/2}} + \frac{72MR^3}{(T+3)^3}.$$

Now, we bound expectation of all error terms. Firstly, we consider $err_{T+1}^{up}$

$$A_T \mathbb{E}\left[err_{T+1}^{up}\right] \overset{(22)}{=} A_T \mathbb{E}\left[\sum_{j=0}^{T} \frac{\alpha_j}{A_j}\langle g(x_{j+1}) - \nabla f(x_{j+1}), x^* - x_{j+1}\rangle\right] = 0.$$

Next, we bound $A_T \mathbb{E}\left[err_{T+1}^v\right]$

$$A_T \mathbb{E}\left[err_{T+1}^v\right] \overset{(35)}{=} A_T \mathbb{E}\left[\sum_{j=0}^{T} \frac{2}{A_j \bar{\delta}_j}\|g(v_j) - \nabla f(v_j)\|^2\right] \leq 2\sigma^2 \sum_{j=0}^{T} \frac{A_T}{A_j \bar{\delta}_j}$$

$$\overset{(44),(48)}{=} \frac{2\sigma R}{3^{3/2}}\sum_{j=0}^{T} \frac{A_T \alpha_j^{3/2}}{A_j} \overset{(46)}{\leq} \frac{2\sigma R}{(T+3)^{1/2}}$$

Now we calculate $A_T \mathbb{E}\left[err_{T+1}^x\right]$

$$A_T \mathbb{E}\left[err_{T+1}^x\right] \overset{(37)}{=} A_T \mathbb{E}\left[\sum_{j=0}^{T} \frac{\alpha_j^2}{2A_j^2 \lambda_j}\|g(x_{j+1}) - \nabla f(x_{j+1})\|^2 + \sum_{j=0}^{T} \frac{\alpha_j}{A_j}\langle g(x_{j+1}) - \nabla f(x_{j+1}), y_j - x_{j+1}\rangle\right]$$

$$\overset{(49)}{=} \frac{\sigma R}{2} \sum_{j=0}^{T} \frac{A_T \alpha_j^2}{A_j^2 (j+3)^{5/2}} \overset{(44)}{=} \frac{\sigma R}{3^{5/2} 2} \sum_{j=0}^{T} \frac{A_T \alpha_j^{9/2}}{A_j^2} \leq \frac{\sigma R}{3^{5/2} 2 A_T} \sum_{j=0}^{T} \frac{A_T \alpha_j^{9/2}}{A_j}$$

$$\overset{(46)}{\leq} \frac{3\sigma R}{2 A_T (T+3)^{7/2}} \overset{(45)}{\leq} \frac{\sigma R}{4(T+3)^{1/2}}.$$

Finally, we consider $err_{T+1}^\tau$

$$A_T \mathbb{E}\left[ err_{T+1}^\tau \right] \overset{(60)}{=} \sum_{j=0}^{T} \frac{A_T \tau^2}{A_j \bar{\delta}_j} \overset{(48)}{=} \frac{\tau R}{3^{3/2}} \sum \frac{A_T \alpha_t^{3/2}}{A_j} \overset{(46)}{\leq} \frac{\tau R}{(T+3)^{1/2}}.$$

Combining all bounds from above we achieve convergence rate

$$\mathbb{E}\left[ f(x_{T+1}) - f(x^*) \right] \leq \frac{19\tau R}{(T+3)^{1/2}} + \frac{27\sigma R}{(T+3)^{1/2}} + \frac{18\delta_2 R^2}{(T+3)^2} + \frac{72 M R^3}{(T+3)^3}.$$

$\square$

The case of stochastic Hessian (Theorem 3.2 under Assumption 2.3) can be obtained in the same way by taking expectation in Lemma E.3.

## F    PROOF OF THEOREM 5.6

Algorithm 2 is the tensor generalization of Algorithm 1. So, we will follow the same steps as in Appendix E.

First of all, we provide tensor counterpart of Lemmas E.1, E.2, E.3. The proofs directly follow the proofs of Lemmas for 2-nd order case.

**Lemma F.1.** *For convex function $f(x)$ and $\psi_t(x)$, we have*

$$\psi_t(x^*) \leq \frac{f(x^*)}{A_{t-1}} + \frac{\bar{\kappa}_2^t + \lambda_t}{2} \|x^* - x_0\|^2 + \sum_{i=3}^{p+1} \frac{\bar{\kappa}_i^t}{i!} \|x^* - x_0\|^i + err_t^{up}, \tag{50}$$

*where*

$$err_t^{up} = \sum_{j=0}^{t-1} \frac{\alpha_j}{A_j} \langle g(x_{j+1}) - \nabla f(x_{j+1}), x^* - x_{j+1} \rangle \tag{51}$$

**Lemma F.2.** *For the function $f(x)$ with $L_p$-Lipschitz-continuous Hessian and $G_i(x_t)$ is $\delta_i^t$-inexact $i$-th order derivative for $v_t, x_{t+1} \in \mathbb{R}^d$ we have*

$$\|\nabla \phi_{v_t}(x_{t+1}) - \nabla f(x_{t+1})\| \leq \sum_{i=2}^{p} \frac{\delta_i^t}{(i-1)!} \|x_{t+1} - v_t\|^i + \frac{L_p}{p!} \|x_{t+1} - v_t\|^{p+1} + \|g(v_t) - \nabla f(v_t)\|,$$
$$\tag{52}$$

*where we denote $\delta_2^t = \delta_t^{v_t, x_{t+1}}$ to simplify the notation.*

**Lemma F.3.** *Let $\{x_t, v_t\}_{t \geq 1}$ be generated by Algorithm 1. Then, for any $\bar{\delta}_t \geq 4\delta_2^t$, $\eta_i \geq 4$, $M \geq \frac{2}{p} L_p$ and $x_{t+1} = S_{M, \bar{\delta}_t}(v_t)$, the following holds*

$$\frac{\tau^2}{\delta_t} + \frac{2}{\delta_t} \|g(v_t) - \nabla f(v_t)\|^2 + \langle \nabla f(x_{t+1}), v_t - x_{t+1} \rangle$$

$$\geq \min \left\{ \frac{2}{p} \|\nabla f(x_{t+1})\|^{\frac{p+1}{p}} \left( \frac{(p-1)!}{M} \right)^{\frac{1}{p}}; \; \frac{1}{4\bar{\delta}_t} \|\nabla f(x_{t+1})\|^2; \right. \tag{53}$$

$$\left. \min_{i=3,\ldots,p} \left( \frac{2}{p} \|\nabla f(x_{t+1})\|^{\frac{i}{i-1}} \left( \frac{(i-1)!}{\eta_i \delta_i^t} \right)^{\frac{1}{i-1}} \right) \right\}.$$

*Proof.* For simplicity, we denote $r_{t+1} = \|x_{t+1} - v_t\|$ and

$$\zeta_{t+1} = \bar{\delta}_t + \sum_{i=3}^{p} \frac{\eta_i \delta_i}{(i-1)!} \|x_{t+1} - v_t\|^{i-2} + \frac{M}{(p-1)!} \|x_{t+1} - v_t\|^{p-1}. \tag{54}$$

By Definition 5.5 for $x_{t+1} = S_p^{M,\bar{\delta}_t,\tau}(v_t)$

$$
\begin{aligned}
\tau^2 &\geq \left\| \nabla\phi_{v_t,p}(x_{t+1}) + \bar{\delta}_t(x_{t+1} - v_t) + \sum_{i=3}^p \frac{\eta_i \delta_i^t}{(i-1)!}\|x_{t+1} - v_t\|^{i-2}(x_{t+1} - v_t) \right. \\
&\qquad \left. + \frac{M}{(p-1)!}\|x_{t+1} - v_t\|^{p-1}(x_{t+1} - v_t) \right\|^2 \\
&\overset{(54)}{=} \|\nabla\phi_{v_t,p} + \zeta_{t+1}(x_{t+1} - v_t)\|^2.
\end{aligned}
\tag{55}
$$

From inexact solution of subproblem we get
$$
\begin{aligned}
\|\nabla f(x_{t+1}) &+ \zeta_{t+1}(x_{t+1} - v_t)\|^2 \\
&\leq \|\nabla\phi_{v_t,p}(x_{t+1}) - \nabla f(x_{t+1}) - \nabla\phi_{v_t,p}(x_{t+1}) - \zeta_{t+1}(x_{t+1} - v_t)\|^2 \\
&\leq 2\|\nabla\phi_{v_t,p}(x_{t+1}) - \nabla f(x_{t+1})\|^2 + 2\|\nabla\phi_{v_t,p}(x_{t+1}) + \zeta_{t+1}(x_{t+1} - v_t)\|^2 \\
&\leq 2\|\nabla\phi_{v_t,p}(x_{t+1}) - \nabla f(x_{t+1})\|^2 + 2\tau^2.
\end{aligned}
\tag{56}
$$

Next, from previous inequality and Lemma F.2
$$
\begin{aligned}
4\|g(v_t) - \nabla f(v_t)\|^2 &+ 4\left( \sum_{i=2}^p \frac{\delta_i^t}{(i-1)!}r_{t+1}^i + \frac{L_p}{p!}r_{t+1}^{p+1}\right)^2 + 2\tau^2 \\
&\geq 2\|\nabla\phi_{v_t,p}(x_{t+1} - v_t)\|^2 + 2\tau^2 \\
&\geq \|\nabla f(x_{t+1}) + \zeta_{t+1}(x_{t+1} - v_t)\|^2 \\
&\geq \|\nabla f(x_{t+1})\|^2 + 2\zeta_{t+1}\langle\nabla f(x_{t+1}), x_{t+1} - v_t\rangle + \zeta_{t+1}^2\|x_{t+1} - v_t\|^2.
\end{aligned}
\tag{57}
$$

Hence,
$$
\frac{\tau^2}{\zeta_{t+1}} + \frac{2}{\zeta_{t+1}}\|g(v_t) - \nabla f(v_t)\|^2 + \langle\nabla f(x_{t+1}), v_t - x_{t+1}\rangle \geq
$$
$$
\frac{1}{2\zeta_{t+1}}\|\nabla f(x_{t+1})\|^2 + \frac{1}{2\zeta_{t+1}}\left( \zeta_{t+1}^2 - 4\left(\sum_{i=2}^p \frac{\delta_i^t}{(i-1)!}r_{t+1}^{i-2} + \frac{L_p}{p!}r_{t+1}^{p-1}\right)^2\right)r_{t+1}^2,
$$
and finally by using definition of $\zeta_{t+1}$, we get
$$
\frac{\tau^2}{\bar{\delta}_t} + \frac{2}{\bar{\delta}_t}\|g(v_t) - \nabla f(v_t)\|^2 + \langle\nabla f(x_{t+1}), v_t - x_{t+1}\rangle \geq
$$
$$
\frac{1}{2\zeta_{t+1}}\|\nabla f(x_{t+1})\|^2 + \frac{1}{2\zeta_{t+1}}\left( \zeta_{t+1}^2 - 4\left(\sum_{i=2}^p \frac{\delta_i^t}{(i-1)!}r_{t+1}^{i-2} + \frac{L_p}{p!}r_{t+1}^{p-1}\right)^2\right)r_{t+1}^2.
$$
Next, we consider $p$ cases depending on which term dominates in $\zeta_{t+1}$.

- If $\bar{\delta}_t \geq \sum\limits_{i=3}^p \frac{\eta_i \delta_i^t}{(i-1)!}r_{t+1}^{i-2} + \frac{M}{(p-1)!}r_{t+1}^{p-1}$, then we get the following bound
$$
\frac{\tau^2}{\bar{\delta}_t} + \frac{2}{\bar{\delta}_t}\|g(v_t) - \nabla f(v_t)\|^2 + \langle\nabla f(x_{t+1}), v_t - x_{t+1}\rangle
$$
$$
\geq \frac{1}{2\zeta_{t+1}}\|\nabla f(x_{t+1})\|^2 + \frac{1}{2\zeta_{t+1}}\left( \zeta_{t+1}^2 - 4\left(\sum_{i=2}^p \frac{\delta_i^t}{(i-1)!}r_{t+1}^{i-2} + \frac{L_p}{p!}r_{t+1}^{p-1}\right)^2\right)r_{t+1}^2
$$
$$
\geq \tfrac{1}{2p\bar{\delta}_t}\|\nabla f(x_{t+1})\|^2.
$$

- If $\frac{\eta_i \delta_i^t}{(i-1)!}r_{t+1}^{i-1} \geq \bar{\delta}_t + \sum\limits_{j=3,j\neq i}^p \frac{\eta_j \delta_j}{(j-1)!}r_{t+1}^{j-1} + \frac{M}{(p-1)!}r_{t+1}^{p-1}$ for $i$, $3 \leq i \leq p$, we get
$$
\frac{\tau^2}{\bar{\delta}_t} + \frac{2}{\bar{\delta}_t}\|g(v_t) - \nabla f(v_t)\|^2 + \langle\nabla f(x_{t+1}), v_t - x_{t+1}\rangle
$$
$$
\geq \frac{1}{2\zeta_{t+1}}\|\nabla f(x_{t+1})\|^2 + \frac{1}{2\zeta_{t+1}}\left( \zeta_{t+1}^2 - 4\left(\sum_{i=2}^p \frac{\delta_i^t}{(i-1)!}r_{t+1}^i + \frac{L_p}{p!}r_{t+1}^{p+1}\right)^2\right)r_{t+1}^2
$$

$$\geq \frac{(i-1)!\|\nabla f(x_{t+1})\|^2}{p\eta_i\delta_i^t r_{t+1}^{i-1}} + \left(\bar{\delta}_t - 2\delta_2^t + \sum_{i=3}^{p} \frac{\eta_i\delta_i^t - 2\delta_i^t}{(i-1)!}r_{t+1}^{i-2} + \frac{pM - 2L_p}{p!}r_{t+1}^{p-1}\right)$$

$$\times \left(\bar{\delta}_t + 2\delta_2^t + \sum_{i=3}^{p} \frac{\eta_i\delta_i^t + 2\delta_i^t}{(i-1)!}r_{t+1}^{i-2} + \frac{pM + 2L_p}{p!}r_{t+1}^{p-1}\right) \frac{r_{t+1}^2}{\zeta_{t+1}}$$

$$\geq \frac{(i-1)!\|\nabla f(x_{t+1})\|^2}{p\eta_i\delta_i^t r_{t+1}^{i-2}} + \frac{\eta_i\delta_i^t - 2\delta_i^t}{(i-1)!}\frac{\eta_i\delta_i^t + 2\delta_i^t}{(i-1)!}\frac{r_{t+1}^i(i-1)!}{p\eta_i\delta_i^t}$$

$$\geq \frac{(i-1)!\|\nabla f(x_{t+1})\|^2}{p\eta_i\delta_i^t r_{t+1}^{i-2}} + \frac{\eta_i\delta_i^t}{(i-1)!p}r_{t+1}^i.$$

$$\geq \frac{2}{i}\left(\frac{(i-2)(i-1)!\|\nabla f(x_{t+1})\|^2}{p\eta_i\delta_i^t}\right)^{\frac{i}{2(i-1)}}\left(\frac{i\eta_i\delta_i^t}{(i-1)!p}\right)^{\frac{i-2}{2(i-1)}}$$

$$\geq \frac{2}{p}\|\nabla f(x_{t+1})\|^{\frac{i}{i-1}}\left(\frac{(i-1)!}{\eta_i\delta_i^t}\right)^{\frac{1}{i-1}},$$

where we used $\frac{\alpha}{(i-2)r^{i-2}} + \frac{\beta r^i}{i} \geq \frac{2}{i}\alpha^{\frac{i}{2(i-1)}}\beta^{\frac{i-2}{2(i-1)}}$.

- If $\frac{M}{(p-1)!}r_{t+1}^{p-1} \geq \bar{\delta}_t + \sum_{i=3}^{p}\frac{\eta_i\delta_i^t}{(i-1)!}r_{t+1}^{i-1}$, then similarly to previous case, we get

$$\frac{\tau^2}{\bar{\delta}_t} + \frac{2}{\bar{\delta}_t}\|g(v_t) - \nabla f(v_t)\|^2 + \langle \nabla f(x_{t+1}), v_t - x_{t+1}\rangle$$

$$\geq \frac{1}{2\zeta_{t+1}}\|\nabla f(x_{t+1})\|^2 + \frac{1}{2\zeta_{t+1}}\left(\zeta_{t+1}^2 - 4\left(\sum_{i=2}^{p}\frac{\delta_i^t}{(i-1)!}r_{t+1}^i + \frac{L_p}{p!}r_{t+1}^{p+1}\right)^2\right)r_{t+1}^2$$

$$\frac{(p-1)!}{pMr_{t+1}^{p-1}}\|\nabla f(x_{t+1})\|^2 + \frac{pM}{2p(p!)}r_{t+1}^{p+1}$$

$$\geq \frac{2}{p}\|\nabla f(x_{t+1})\|^{\frac{p+1}{p}}\left(\frac{(p-1)!}{M}\right)^{\frac{1}{p}}.$$

$\square$

Next we will need technical Lemmas E.4, E.5.

**Lemma F.4.** *Let $\{x_t, y_t\}_{t\geq 1}$ be generated by Algorithm 2. Then*

$$\psi_t^* = \min_x \psi_t(x) \geq \frac{f(x_t)}{A_{t-1}} - err_t^v - err_t^x - err_t^\tau, \tag{58}$$

*where*

$$err_t^v = \sum_{j=0}^{t-1}\frac{2}{A_j\bar{\delta}_j}\|g(v_j) - \nabla f(v_j)\|^2, \tag{59}$$

$$err_t^\tau = \sum_{j=0}^{t-1}\frac{\tau^2}{A_j\bar{\delta}_j}, \tag{60}$$

*and*

$$err_t^x = \sum_{j=0}^{t-1}\frac{\alpha_j^2}{2A_j^2\lambda_j}\|g(x_{j+1}) - \nabla f(x_{j+1})\|^2 + \sum_{j=0}^{t-1}\frac{\alpha_j}{A_j}\langle g(x_{j+1}) - \nabla f(x_{j+1}), y_j - x_{j+1}\rangle. \tag{61}$$

*Proof.* We prove Lemma by induction. Let us start with $t = 0$, we define $A_{-1}$ such that $\frac{1}{A_{-1}} = 0$. Then $\frac{f(x_0)}{A_{-1}} = 0$ and $\psi_0^* = 0$, hence, $\psi_0^* \geq \frac{f(x_0)}{A_{-1}}$. Let us assume that (58) is true for $t$ and show that (58) is true for $t + 1$.

Following the steps of Lemma E.6 for $p$-th order case, we get

$$\psi_{t+1}^* = \psi_{t+1}(y_{t+1}) \geq \frac{f(x_{t+1})}{A_t} + \frac{1}{A_t}\langle\nabla f(x_{t+1}), v_t - x_{t+1}\rangle$$

$$+ \sum_{i=2}^{p+1}\left(\frac{1}{2}\right)^{i-2}\frac{\bar{\kappa}_i^t}{(i-1)!}d_i(y_{t+1} - y_t) + \frac{\alpha_t}{A_t}\langle\nabla f(x_{t+1}), y_{t+1} - y_t\rangle - err_t^v - err_{t+1}^x - err_t^\tau. \tag{62}$$

To complete the induction step we will show, that the sum of all terms in the RHS except $\frac{f(x_{t+1})}{A_t}$ and error terms is non-negative.

Lemma F.3 provides the lower bound for $\langle\nabla f(x_{t+1}), v_t - x_{t+1}\rangle$. Let us consider the case when the minimum in the RHS of (53) is attained at the term with particular $i = 3, \ldots, p$. By Lemma E.5 with the following choice of the parameters

$$z = y_t - y_{t+1}, \quad v_t = \frac{\alpha_t}{A_t}\nabla f(x_{t+1}), \quad \theta = \left(\frac{1}{2}\right)^{i-2}\frac{\bar{\kappa}_i^t}{(i-1)!},$$

we have

$$\frac{1}{i}\left(\frac{1}{2}\right)^{i-2}\frac{\bar{\kappa}_i^t}{(i-1)!}\|y_t - y_{t+1}\|^i + \frac{\alpha_t}{A_t}\langle\nabla f(x_{t+1}), y_{t+1} - y_t\rangle \geq -\frac{i-1}{i}\left(\frac{\|\frac{\alpha_t}{A_t}\nabla f(x_{t+1})\|^i}{\left(\frac{1}{2}\right)^{i-2}\frac{\bar{\kappa}_i^t}{(i-1)!}}\right)^{\frac{1}{i-1}}. \tag{63}$$

Hence,

$$\frac{f(x_{t+1})}{A_t} + \frac{1}{A_t}\langle\nabla f(x_{t+1}), v_t - x_{t+1}\rangle + \left(\frac{1}{2}\right)^{i-2}\frac{\bar{\kappa}_i^t}{(i-1)!}d_i(y_{t+1} - y_t) + \frac{\alpha_t}{A_t}\langle\nabla f(x_{t+1}), y_{t+1} - y_t\rangle$$

$$\overset{(39)}{\geq} \frac{f(x_{t+1})}{A_t} + \frac{1}{A_t}\langle\nabla f(x_{t+1}), v_t - x_{t+1}\rangle - \frac{i-1}{i}\left(\frac{\|\frac{\alpha_t}{A_t}\nabla f(x_{t+1})\|^i}{\left(\frac{1}{2}\right)^{i-2}\frac{\bar{\kappa}_i^t}{(i-1)!}}\right)^{\frac{1}{i-1}}$$

$$\overset{(53)}{\geq} \frac{f(x_{t+1})}{A_t} + \frac{2}{p}\|\nabla f(x_{t+1})\|^{\frac{i}{i-1}}\left(\frac{(i-1)!}{\eta_i\delta_i^t}\right)^{\frac{1}{i-1}} - \frac{i-1}{i}\left(\frac{\|\frac{\alpha_t}{A_t}\nabla f(x_{t+1})\|^i}{\left(\frac{1}{2}\right)^{i-2}\frac{\bar{\kappa}_i^t}{(i-1)!}}\right)^{\frac{1}{i-1}}$$

$$\geq \frac{f(x_{t+1})}{A_t},$$

where the last inequality holds by our choice of the parameters

$$\bar{\kappa}_i^t \geq \frac{p^{i-1}}{2}\frac{\alpha_t^i}{A_t}\eta_i\delta_i^t. \tag{64}$$

Let us consider the case when the minimum in the RHS of (53) is achieved on the second term. Following similar steps, we get

$$\bar{\kappa}_2^t \geq \frac{2p\alpha_t^2}{A_t}\bar{\delta}_t. \tag{65}$$

Next, we consider the case when the minimum in the RHS of (53) is achieved on the first term. Again, by Lemma E.5 with the same choice of $z, v$ and with $\theta = \left(\frac{1}{2}\right)^{p-1}\frac{\bar{\kappa}_{p+1}^{t-1}}{(p-1)!}$, we have

$$\frac{1}{p}\left(\frac{1}{2}\right)^{p-1}\frac{\bar{\kappa}_{p+1}^{t-1}}{p!}\|y_t - y_{t+1}\|^{p+1} + \frac{\alpha_t}{A_t}\langle\nabla f(x_{t+1}), y_{t+1} - y_t\rangle \geq -\frac{p}{p+1}\left(\frac{\|\frac{\alpha_t}{A_t}\nabla f(x_{t+1})\|^{p+1}}{\left(\frac{1}{2}\right)^{p-1}\frac{\bar{\kappa}_{p+1}^t}{(p-1)!}}\right)^{\frac{1}{p}}. \tag{66}$$

Hence, we get

$$\frac{f(x_{t+1})}{A_t} + \frac{1}{A_t}\langle\nabla f(x_{t+1}), v_t - x_{t+1}\rangle + \frac{1}{2^{p-1}}\frac{\bar{\kappa}_{p+1}^{t-1}}{p!}d_{p+1}(y_{t+1} - y_t) + \frac{\alpha_t}{A_t}\langle\nabla f(x_{t+1}), y_{t+1} - y_t\rangle$$

$$\overset{(66)}{\geq} \frac{f(x_{t+1})}{A_t} + \frac{1}{A_t}\langle\nabla f(x_{t+1}), v_t - x_{t+1}\rangle - \frac{p}{p+1}\left(\frac{\|\frac{\alpha_t}{A_t}\nabla f(x_{t+1})\|^{p+1}}{\left(\frac{1}{2}\right)^{p-1}\frac{\bar{\kappa}_{p+1}^t}{p!}}\right)^{\frac{1}{p}}$$

$$\overset{(53)}{\geq} \frac{f(x_{t+1})}{A_t} + \frac{1}{A_t}\frac{2}{p}\|\nabla f(x_{t+1})\|^{\frac{p+1}{p}}\left(\frac{(p-1)!}{M}\right)^{\frac{1}{p}} - \frac{p}{p+1}\left(\frac{\|\frac{\alpha_t}{A_t}\nabla f(x_{t+1})\|^{p+1}}{\left(\frac{1}{2}\right)^{p-1}\frac{\bar{\kappa}_{p+1}^t}{p!}}\right)^{\frac{1}{p}}$$

$$\geq \frac{f(x_{t+1})}{A_t},$$

where the last inequality holds by our choice of $\bar{\kappa}_{p+1}^t$:

$$\bar{\kappa}_{p+1}^t \geq \frac{(p+1)^{p+1}}{2}\frac{\alpha_t^{p+1}}{A_t}M. \tag{67}$$

To sum up, by our choice of the parameters $\bar{\kappa}_i^t$, $i = 2, ..., p$, we obtain

$$\psi_{t+1}^* \geq \frac{f(x_{t+1})}{A_t} - err_{t+1}^v - err_{t+1}^x - err_{t+1}^\tau.$$

$\square$

**Theorem F.5.** *Let Assumption 5.1 hold and $M \geq \frac{2}{p}L_p$. Let Assumption 5.4 hold. After $T \geq 1$ with parameters defined in (17) and $\sigma_2 = \delta_2 = \max\limits_{t=1,...,T}\delta_t^{v_{t-1},x_t}$ we get the following bound for the objective residual*

$$\mathbb{E}\left[f(x_T) - f(x^*)\right] \leq \frac{2(p+1)^3\tau R}{(T+p+1)^{1/2}} + \frac{3(p+1)^3\sigma R}{(T+p+1)^{1/2}} + \sum_{i=3}^p \frac{2(p+1)^{2i-1}\delta_i R^i}{i!(T+p+1)^i} + \frac{(p+1)^{2(p+1)}}{(p+1)!}\frac{MR^{p+1}}{(T+p+1)^{p+1}}$$

*Proof.* First of all, let us bound $A_T$.

$$\alpha_t = \frac{p+1}{t+p+1}, \ t \geq 1. \tag{68}$$

Then, we have

$$\frac{1}{(T+p+1)^{p+1}} \leq A_T \leq \frac{(p+1)!}{(T+1)^{p+1}}. \tag{69}$$

And from Agafonov et al. (2023) we get

$$\sum_{t=0}^T \frac{A_T\alpha_t^i}{A_t} \leq \frac{(p+1)^i}{(T+p+1)^{i-1}} \tag{70}$$

From Lemmas F.4 and F.1, we obtain that, for all $t \geq 1$,

$$\frac{f(x_{t+1})}{A_t} - err_{t+1}^v - err_{t+1}^x - err_{t+1}^\tau \overset{(58)}{\leq} \psi_{t+1}^* \leq \psi_{t+1}(x^*)$$

$$\overset{(50)}{\leq} \frac{f(x^*)}{A_t} + \frac{\bar{\kappa}_2^t + \lambda_t}{2}\|x^* - x_0\|^2 + \frac{\bar{\kappa}_3^t}{6}\|x^* - x_0\|^3 + err_{t+1}^{up}.$$

Next, we apply expectation

$$\mathbb{E}\left[f(x_{T+1}) - f(x^*)\right] \leq A_T\mathbb{E}\left[\frac{\bar{\kappa}_2^T + \lambda_T}{2}R^2 + \sum_{i=3}^{p+1}\frac{\bar{\kappa}_i^T}{i!}R^i + err_{T+1}^{up} + err_{T+1}^v + err_{T+1}^x + err_{T+1}^\tau\right]. \tag{71}$$

Let us choose

$$\bar{\delta}_t = \delta_2 + \frac{\tau + \sigma}{R}(t+p+1)^{3/2}, \tag{72}$$

$$\lambda_t = \frac{\sigma}{R}(t+p+1)^{p+1/2}. \tag{73}$$

Then, we bound terms in (71) step by step.

We start from deterministic terms.

$$A_T \mathbb{E}\left[\frac{\bar{\kappa}_2^T + \lambda_T}{2}R^2 + \sum_{i=3}^{p}\frac{\bar{\kappa}_i^T}{i!}R^i + \frac{\bar{\kappa}_{p+1}^T}{(p+1)!}R^{p+1}\right]$$

$$\overset{(17)}{=} p\alpha_T^2\bar{\delta}_T R^2 + \sum_{i=3}^{p}\frac{4p^{i-1}}{i!2}\alpha_T^i\delta_i R^i + \frac{(p+1)^{p+1}}{(p+1)!}\alpha_T^{p+1}MR^{p+1}$$

$$\overset{(72),(73)}{=} \frac{(p+1)^3(\tau+\sigma)R}{(T+p+1)^{1/2}} + \frac{(p+1)^3\delta_2 R^2}{(T+p+1)^2} + \sum_{i=3}^{p}\frac{2(p+1)^{2i-1}\delta_i R^i}{i!(T+p+1)^i} + \frac{(p+1)^{2(p+1)}}{(p+1)!}\frac{MR^{p+1}}{(T+p+1)^{p+1}}.$$

Now, we bound expectation of all error terms. Firstly, we consider $err_{T+1}^{up}$

$$A_T\mathbb{E}\left[err_{T+1}^{up}\right] = A_T\mathbb{E}\left[\sum_{j=0}^{T}\frac{\alpha_j}{A_j}\langle g(x_{j+1}) - \nabla f(x_{j+1}), x^* - x_{j+1}\rangle\right] = 0.$$

Next, we bound $A_T\mathbb{E}\left[err_{T+1}^{v}\right]$

$$A_T\mathbb{E}\left[err_{T+1}^{v}\right] = A_T\mathbb{E}\left[\sum_{j=0}^{T}\frac{2}{A_j\bar{\delta}_j}\|g(v_j) - \nabla f(v_j)\|^2\right] \leq 2\sigma^2\sum_{j=0}^{T}\frac{A_T}{A_j\bar{\delta}_j}$$

$$\overset{(68),(72)}{=} \frac{2\sigma R}{(p+1)^{3/2}}\sum_{j=0}^{T}\frac{A_T\alpha_j^{3/2}}{A_j} \overset{(70)}{\leq} \frac{2\sigma R}{(T+p+1)^{1/2}}$$

Now we calculate $A_T\mathbb{E}\left[err_{T+1}^{x}\right]$

$$A_T\mathbb{E}\left[err_{T+1}^{x}\right] \overset{(37)}{=} A_T\mathbb{E}\left[\sum_{j=0}^{T}\frac{\alpha_j^2}{2A_j^2\lambda_j}\|g(x_{j+1}) - \nabla f(x_{j+1})\|^2 + \sum_{j=0}^{T}\frac{\alpha_j}{A_j}\langle g(x_{j+1}) - \nabla f(x_{j+1}), y_j - x_{j+1}\rangle\right]$$

$$\overset{(73)}{=} \frac{\sigma R}{2}\sum_{j=0}^{T}\frac{A_T\alpha_j^2}{A_j^2(j+3)^{p+1/2}} \overset{(44)}{=} \frac{\sigma R}{(p+1)^{5/2}2}\sum_{j=0}^{T}\frac{A_T\alpha_j^{p+5/2}}{A_j^2} \leq \frac{\sigma R}{(p+1)^{5/2}2A_T}\sum_{j=0}^{T}\frac{A_T\alpha_j^{p+5/2}}{A_j}$$

$$\overset{(46)}{\leq} \frac{(p+1)^2\sigma R}{2A_T(T+p+1)^{p+3/2}} \overset{(45)}{\leq} \frac{(p+1)^2\sigma R}{2(T+p+1)^{1/2}}.$$

Finally, we consider $err_{T+1}^{\tau}$

$$A_T\mathbb{E}\left[err_{T+1}^{\tau}\right] = \sum_{j=0}^{T}\frac{A_T\tau^2}{A_j\bar{\delta}_j} \overset{(72)}{=} \frac{\tau R}{(p+1)^{3/2}}\sum\frac{A_T\alpha_t^{3/2}}{A_j} \overset{(46)}{\leq} \frac{\tau R}{(T+p+1)^{1/2}}.$$

Combining all bounds from above we achieve convergence rate

$$\mathbb{E}\left[f(x_{T+1}) - f(x^*)\right] \leq \frac{2(p+1)^3\tau R}{(T+p+1)^{1/2}} + \frac{3(p+1)^3\sigma R}{(T+p+1)^{1/2}} + \sum_{i=3}^{p}\frac{2(p+1)^{2i-1}\delta_i R^i}{i!(T+p+1)^i} + \frac{(p+1)^{2(p+1)}}{(p+1)!}\frac{MR^{p+1}}{(T+p+1)^{p+1}}.$$

$\square$

Again, the case of stochastic Hessian (Theorem 5.6 under Assumption 5.3) can be obtained in the same way by taking expectation in Lemma F.3.

# G   RESTARTS FOR STRONGLY-CONVEX FUNCTION

**Theorem 6.2.** *Let Assumption 6.1 hold and let parameters of Algorithm 1 be chosen as in* (17). *Let* $\{z_s\}_{s\geq 0}$ *be generated by Algorithm 3 and* $R > 0$ *be such that* $\|z_0 - x^*\| \leq R$. *Then for any* $s \geq 0$ *we have*

$$\mathbb{E}\|z_s - x^*\|^2 \leq 4^{-s}R^2, \tag{74}$$

$$\mathbb{E}f(z_s) - f(x^*) \leq 2^{-2s-1}\mu R^2. \tag{75}$$

*Moreover, the total number of iterations to reach desired accuracy $\varepsilon$ : $f(z_s) - f(x^*) \leq \varepsilon$ in expectation is*

$$O\left(\frac{(\tau+\sigma_1)^2}{\mu\varepsilon} + \left(\sqrt{\frac{\sigma_2}{\mu}} + 1\right)\log\frac{f(z_0)-f(x^*)}{\varepsilon} + \sum_{i=3}^{p}\left(\frac{\sigma_i R^{i-2}}{\mu}\right)^{\frac{1}{i}} + \left(\frac{L_p R^{p-1}}{\mu}\right)^{\frac{1}{p+1}}\right). \tag{76}$$

*Proof.* We prove by induction that $\mathbb{E}\|z_s - x^*\|^2 \leq 4^{-s}\|z_0 - x^*\|^2 = 4^{-s}R_0^2$. For $s = 0$ this obviously holds. By strong convexity and convergence of Algorithm 2

$$\mathbb{E}\left[f(x_T) - f(x^*)\right] \leq \frac{C_\tau \tau R}{\sqrt{T}} + \frac{C_1\sigma_1 R}{\sqrt{T}} + \sum_{i=2}^{p}\frac{C_i\sigma_i R^i}{T^i} + \frac{C_{p+1}L_p R^{p+1}}{T^{p+1}}$$

we get

$$\mathbb{E}_{[z_{s+1}|z_s,z_{s-1},...,z_0]}\|z_{s+1} - x^*\|^2 \leq \frac{2}{\mu}\mathbb{E}_{[z_{s+1}|z_s,...,z_0]}(f(x_{t_s+1}) - f(x^*))$$

$$\leq \frac{2}{\mu}\left(\frac{C_\tau \tau R}{\sqrt{T}} + \frac{C_1\sigma_1 R}{\sqrt{T}} + \sum_{i=2}^{p}\frac{C_i\sigma_i R^i}{T^i} + \frac{C_{p+1}L_p R^{p+1}}{T^{p+1}}.\right)$$

$$\leq \frac{2}{\mu}\left(\frac{\mu C_\tau \sigma\|z_s - x^*\|}{8(p+2)C_\tau\sigma\|z_s - x^*\|} + \frac{\mu C_1\sigma\|z_s - x^*\|}{8(p+2)C_1\sigma\|z_s - x^*\|}\right.$$

$$+ \sum_{i=2}^{p}\frac{\mu C_i\delta_i\|z_s - x^*\|^i}{8(p+2)C_i\delta_i\|z_s - x^*\|^i} + \left.\frac{\mu C_{p+1}L_p\|z_s - x^*\|^{p+1}}{8(p+2)C_{p+1}L_p\|z_s - x^*\|^{p+1}}\right)$$

$$\leq \frac{p}{4(p+2)}\|z_s - x^*\|^2 + \frac{2}{4(p+2)}r_s\|z_s - x^*\|.$$

Then by taking full expectation we obtain

$$\mathbb{E}\|z_{s+1} - x^*\|^2 \leq \frac{p}{4(p+1)}\mathbb{E}\|z_s - x^*\|^2 + \frac{2}{4(p+2)}r_s\mathbb{E}\|z_s - x^*\| \leq \frac{1}{4^{s+1}}R_0^2$$

Thus, by induction, we obtain that (74), (75) hold.

Next we provide the corresponding complexity bounds. From the above induction bounds, we obtain that after $S$ restarts the total number of iterations of Algorithm 2

$$\mathbb{E}T = \mathbb{E}\sum_{s=1}^{S}t_s \leq \sum_{s=1}^{S}\max\left\{1, \left(\frac{8(p+2)C_\tau\tau}{\mu r_{s-1}}\right)^2, \left(\frac{8(p+2)C_1\sigma_1}{\mu r_{s-1}}\right)^2, \max_{i=1,...,p}\left(\frac{8(p+2)C_i\delta_i R_{s-1}^{i-2}}{\mu}\right)^{\frac{1}{i}}, \left(\frac{8(p+2)C_{p+1}L_p R_{s-1}^{p-1}}{\mu}\right)^{\frac{1}{p+1}}\right\}$$

$$\leq \sum_{s=1}^{S}\left(1 + \left(\frac{8(p+2)C_\tau\tau}{\mu r_{s-1}}\right)^2 + \left(\frac{8(p+2)C_1\sigma_1}{\mu r_{s-1}}\right)^2 + \sum_{i=2}^{p}\left(\frac{8(p+2)C_i\delta_i R_{s-1}^{i-2}}{\mu}\right)^{\frac{1}{i}} + \left(\frac{8(p+2)C_{p+1}L_p R_{s-1}^{p-1}}{\mu}\right)^{\frac{1}{p+1}}\right)$$

$$\leq S + \left(\frac{8(p+2)C_\tau\tau}{\mu R_0}\right)^2 4^S + \left(\frac{8(p+2)C_1\sigma_1}{\mu R_0}\right)^2 4^S + \left(\frac{8(p+2)C_2\delta_2}{\mu}\right)^{\frac{1}{2}}S + \sum_{i=3}^{p}\left(\frac{8(p+2)C_i\delta_i R_0^{i-2}}{\mu}\right)^{\frac{1}{i}} + \left(\frac{8(p+2)C_{p+1}L_p R_0^{p-1}}{\mu}\right)^{\frac{1}{p+1}}$$

$$\leq \log\frac{f(z_0) - f(x^*)}{\varepsilon} + \left(\frac{8(p+2)\tau}{\mu R_0}\right)^2\frac{f(z_0)-f(x^*)}{\varepsilon} + \left(\frac{8(p+2)C_1\sigma_1}{\mu R_0}\right)^2\frac{f(z_0)-f(x^*)}{\varepsilon}$$

$$+ \left(\frac{8(p+2)C_2\delta_2}{\mu}\right)^{\frac{1}{2}}\log\frac{f(z_0)-f(x^*)}{\varepsilon} + \sum_{i=3}^{p}\left(\frac{8(p+2)C_i\delta_i R_0^{i-2}}{\mu}\right)^{\frac{1}{i}} + \left(\frac{8(p+2)L_p C_{p+1} R_0^{p-1}}{\mu}\right)^{\frac{1}{p+1}}.$$

Therefore, the total oracle complexities are given by (76). $\qquad\square$

For the case of stochastic high-order derivative the proof remains same w.r.t. change $\delta_i$ to $\sigma_i$.

## H ON THE SOLUTION OF SUBPROBLEM (9) IN ALGORITHM 1

The subproblem (9) admits a closed form solution, which we will derive in the following steps. First, note that the problem (9) is convex. Let us write optimality condition

$$0 = \nabla\psi_t(y_t) = (\lambda_t + \bar{\kappa}_2^t + \bar{\kappa}_3^t\|y_t - x_0\|)(y_t - x_0) + \sum_{j=0}^{t-1}\frac{\alpha_j}{A_j}g(x_{j+1}).$$

Thus,

$$(\lambda_t + \bar{\kappa}_2^t + \bar{\kappa}_3^t \|y_t - x_0\|)\|(y_t - x_0)\| = \|\sum_{j=0}^{t-1} \frac{\alpha_j}{A_j} g(x_{j+1})\|.$$

Let us denote $r_t = \|(y_t - x_0)\|$ and $S_t = \|\sum_{j=0}^{t-1} \frac{\alpha_j}{A_j} g(x_{j+1})\|$. Then, we get

$$\bar{\kappa}_3^t r_t^2 + (\lambda_t + \bar{\kappa}_2^t) r_t - S_t = 0.$$

Next, we get the solution of quadratic equation

$$r_t = \frac{\sqrt{(\lambda_t + \bar{\kappa}_2^t)^2 + 4\bar{\kappa}_3^t S_t} - (\lambda_{t+1} + \bar{\kappa}_2^t)}{2\bar{\kappa}_3^t}.$$

Finally, we get explicit solution

$$y_t = x_0 - \frac{\sum_{j=0}^{t-1} \frac{\alpha_j}{A_j} g(x_{j+1})}{\lambda_t + \bar{\kappa}_2^t + \bar{\kappa}_3^t r_t}.$$

We use the explicit solution in our implementation of the Algorithm 1.

