# OpenReview forum: "Advancing the Lower Bounds: an Accelerated, Stochastic, Second-order Method with Optimal Adaptation to Inexactness"
_ICLR.cc/2024/Conference — ICLR 2024 poster_

### Official Review · Reviewer_xECU · 2023-10-28

**Soundness:** 3 good
**Presentation:** 3 good
**Contribution:** 2 fair
**Rating:** 6
**Confidence:** 3

**Summary:**

This work studies the unconstrained minimization of convex and strongly convex functions with continuously differentiable gradients through second-order methods with access to inexact gradients and Hessians. Further with the assumption of unbiased gradients they show a faster convergence rate of their proposed algorithm than the state of the art. Also, they show the worst-case complexity lower bound for their problem. Then, the proposed method is extended to tensor analysis and a restart scheme is proposed for strongly convex functions. Experimental results confirm the superiority of the proposed method to Extra-Newton and SGD methods.

**Strengths:**

1- Good flow in introduction and problem setup.

2- With additional assumption on the unbiasedness of the gradient, they show the lower bound on convergence rate of inexact Hessian & gradient plus proposing a method which achieves the lower bound exept for the last term.

3- They did a relatively thorough analysis by extending their analysis to tensor methods and special cases like strongly convex cases.

**Weaknesses:**

1- Though the text has a good flow in terms of the context, there is still room for improvements: e.g.

(a) In page 2 there are two loose sentences in the middle of the page. They can be integrated with the previous paragraph or just in a new independent paragraph.

(b) Below assumption 1.2: $E[g(x,\xi)]$ or $E[F(x,\xi)]$?

(c) **\citet** was used wrongly in many places. Please consider replacing the wrong ones with **\citep**.

(d) function $\psi_t(x)$ above algorithm 1 is not defined (I think it is in the algorithm so you should refer to that or simply introduce it)

(e) The definition of $f$ in section 7 page 8 is an abuse of notation: $f(x)=E[f(x,\xi)]$

(f) I suggest larger font size on Figures 1,2

(g) Large gap in the appendix after (26)

2- I think contributions 3 & 1 should be integrated.

3- Table 1 might lead to misunderstanding. Your result is based on the asssumption of unbiased gradients. The rate by Agafonov does not have this assumption. Thus, it seems like an unfair comparison.

4- The title of the section “Lower Bound” is very generic and also the description is vague. For example, the first sentence of this section is vague.

**Questions:**

1- Is the $\boldsymbol \Phi_x$ defined at the beginning of Section 2 used? Same question holds for $\boldsymbol\Phi_x$ defined under Assumption 5.1.

2- Lemma 2.1 seems like a special case of a similar lemma in Agafonov et al 2020. Is there a reason you did not cite their work when you present the Lemma? Same question holds for Lemma 5.2.

3- How did you find the dynamic strategy for the precision level $(\tau_t=c/(t^{3/2}))$

4- According to your investigation and results, does it make sense to analyze inexactness in Hessian (or higher order information of the objective function) when gradients are inexact? This question mainly concerns the $O(1/\sqrt{T})$ convergence rate related to the inexactness of the gradients which dominates the convergence rate as $T\rightarrow \infty$.

---

> ### Author Response · Authors · 2023-11-14
>
> We appreciate your valuable input and considerate questions!
>
> >**Weakness 1 :** *“ Though the text has a good flow in terms of the context, there is still room for improvements: e.g.
> (a) In page 2 there are two loose sentences in the middle of the page. They can be integrated with the previous paragraph or just in a new independent paragraph;
> (b) Below assumption 1.2: E[g(x,ξ)] or E[F(x,ξ)];
> (c) \citet was used wrongly in many places. Please consider replacing the wrong ones with \citep;
> (d) function ψt(x) above algorithm 1 is not defined (I think it is in the algorithm so you should refer to that or simply introduce it);
> (e) The definition of f  in section 7 page 8 is an abuse of notation: f(x)=E[f(x,ξ)];
> (f) I suggest larger font size on Figures 1,2;
> (g) Large gap in the appendix after (26);
> ”*
>
> **Response:** Thank you for suggestions and pointing out several inaccuracies. We agree with comments (a, c, d, e, f , g) and will change it in the revised version. Regarding comment (b), we maintain the expression "E[g(x,ξ)]" exactly as it is below Assumption 1.2
>
> >**Weakness 2:** *"I think contributions 3 & 1 should be integrated."*
>
> **Response:**  Contribution 1 signifies an enhanced convergence rate, whereas Contribution 3 underscores the successful integration of an inexact subproblem solution—a novel aspect not previously addressed in inexact second-order and high-order convex methods. We appreciate your thoughtful input and will consider how best to integrate these contributions for an improved presentation of our research.
>
> >**Weakness 3:** *"Table 1 might lead to misunderstanding. Your result is based on the asssumption of unbiased gradients. The rate by Agafonov does not have this assumption. Thus, it seems like an unfair comparison."*
>
> **Response:** Certainly, we acknowledge that a direct comparison of the convergence of these methods is not feasible, as elucidated in the paragraph following Assumption 2.3. To provide a comprehensive overview of the current landscape in the realm of inexact/stochastic second-order methods, we present Table 1. This table includes a column outlining assumptions related to inexactness, explicitly noting that the unbiasedness of the gradient was not assumed in the work [1]. We believe that Table 1's clear articulation of assumptions minimizes any potential misunderstandings regarding the comparative context.
>
> >**Weakness 4:** *"The title of the section “Lower Bound” is very generic and also the description is vague. For example, the first sentence of this section is vague."*
>
> **Response:** Could you kindly offer additional insights on why the title and the first sentence might be considered vague? We value your perspective and would appreciate any suggestions you may have for improvement. While the current wording seems acceptable to us, we are open to refining it based on your feedback. Thank you.
>
> >**Question 1:** *" Is the $\Phi_x$ defined at the beginning of Section 2 used? Same question holds for $\Phi_x$ defined under Assumption 5.1."*
>
> **Response:** Formally, its utilization is confined to the Appendix. Nevertheless, we believe it is crucial to introduce the exact Taylor polynomial $\Phi_x(y)$, as it provides valuable insights into the construction of high-order methods.
>
> >**Question 2:** *"Lemma 2.1 seems like a special case of a similar lemma in Agafonov et al 2020. Is there a reason you did not cite their work when you present the Lemma? Same question holds for Lemma 5.2."*
>
> **Response:**
> Thank you, we agree with your comment, and we will add a direct citation near Lemmas 2.1 and 5.1 in revised version.
>
> >**Question 3:** *"How did you find the dynamic strategy for the precision level $\tau_t = c / t^{3/2}$?"*
>
> **Response:**
> Thank you for this question. We have a typo here, it should be $\tau = \frac{c}{t^{5/2}}$, we will fill it in the revised version.
>
> From the convergence, we have for any $t \geq 1$
>
> $f(x_{t}) - f(x^*) \leq O(\frac{\sigma_1 R}{\sqrt{t}} + \frac{\sigma_2 R^2}{t^2} + \frac{L_2 R^3}{t^3} + \frac{\sigma_1 R}{\sqrt{t}})$
>
> By choosing $\tau = \frac{c}{t^{5/2}}$, we get for any $t \geq 1$
>
> $f(x_{t}) - f(x^*) \leq O(\frac{\sigma_1 R}{\sqrt{t}} + \frac{\sigma_2 R^2}{t^2} + \frac{L_2 R^3}{t^3} + \frac{c R}{t^3}) $
>
> *References*
>
> [1] Agafonov, Artem, et al. "Inexact tensor methods and their application to stochastic convex optimization." arXiv preprint arXiv:2012.15636 (2020)."*

---

> > ### Author Response · Authors · 2023-11-14
> >
> > >**Question 4:** *"According to your investigation and results, does it make sense to analyze inexactness in Hessian (or higher order information of the objective function) when gradients are inexact? This question mainly concerns the O(1/T) convergence rate related to the inexactness of the gradients which dominates the convergence rate as T→∞."*
> >
> > **Response:** From a theoretical perspective, we agree that the rate of $O(\frac{1}{\sqrt{T}})$ is achieved when $T \to \infty$. However, in practice, we are not interested in this case, as we have either predefined number of iterations or a desired accuracy to achieve. In that case the convergence rate of the method depends on the relationship between constants $M,\sigma_1,\sigma_2$ and can be dominated by either term in rates (9-10).
> >
> > We think that there are several reasons why stochastic high-order methods are important:
> > 1. Please have a look at experiments. Although all methods achieve $O(\frac{1}{\sqrt{T}})$ while $T \to \infty$, the superiority of second-order methods is clear (Figure 2). Moreover, even the power in Hessian inexactness convergence term plays an important role in the behavior of the methods (Figure 2 (c, d)).
> > 2. Next, imagine the case when we can control inexactness in gradients and Hessians. Then, with proper choice of $\sigma_1 = O(\varepsilon^{5/6})$, $\sigma_2  =O(\varepsilon^{1/3})$ (see Corollary 3.3) the method achieves convergence rate $T = O(\varepsilon ^{-1/3})$. With proper choice of gradient inexactness gradient-type methods can not be faster than $O(\varepsilon^{-1/2})$. One particular case, where the inexactness can be controlled is finite-sum optimization, where we can choose batch sizes. Then, in the case of the parallelized computations, we care more about the total number of iterations over the total stochastic derivatives computations.
> > 3. Also consider the following scenario: a distributed centralized network where the server stores a data sample $f_1(x)$ with a Hessian $\nabla^2 f_1(x)$ that offers a "good" approximation of the exact Hessian of the objective function:
> > $$ ||\nabla^2 f_1(x) - \nabla^2 f(x) || \leq \beta.$$
> > This assumption, known as beta-similarity (Hessian-similarity) [1], can be inferred from equation (5 with $H(x) = \nabla^2 f_1(x)$ and $\delta_2 = \beta$. By leveraging this approximation, we can avoid exchanging Hessians, resulting in significant communication savings. This allows running on a server a second-order method with only gradient communications.
> >
> > [1] Zhang, Yuchen, and Xiao Lin. "DiSCO: Distributed optimization for self-concordant empirical loss." International conference on machine learning. PMLR, 2015.

---

> > ### Comment · Reviewer_xECU · 2023-11-19
> >
> > Thanks for your response. Regarding weakness 4 some suggestions might be "worst-case analysis" or "Theoretical Complexity Lower Bounds".
> >
> > The term lower bound is used throughout the paper. Maybe just mention at the beginning that for brevity you abbreviate theoretical complexity lower bound with simply lower bound. Other points are clarified, but they were minor issues. Thus, I keep my score unchanged. Thanks!

---

> > > ### Author Response · Authors · 2023-11-20
> > >
> > > Thank you! We will modify the title of Section 4 as suggested. To ensure clarity, we will explicitly use "theoretical complexity lower bound" before introducing the abbreviation "lower bound." These adjustments will be incorporated into the revised version. We appreciate your valuable insights once again!

---

> > > > ### Author Response · Authors · 2023-11-21
> > > >
> > > > Dear Reviewer xECU,
> > > >
> > > > We have submitted a revised version of the paper, with changes highlighted in blue for your convenience. We believe that we have addressed the comments and suggestions you provided.
> > > >
> > > > Thank you once again for your valuable feedback.

---

### Official Review · Reviewer_2pJU · 2023-10-29

**Soundness:** 3 good
**Presentation:** 3 good
**Contribution:** 3 good
**Rating:** 6
**Confidence:** 4

**Summary:**

The paper proposes an accelerated stochastic second-order algorithm using inexact gradients and Hessians, demonstrating nearly optimal convergence rates.

**Strengths:**

The paper proposes an accelerated stochastic second-order algorithm using inexact gradients and Hessians, demonstrating nearly optimal convergence rates.

**Weaknesses:**

see questions.

**Questions:**

1) The paper lacks an intuitive explanation of the proposed algorithm. It is suggested to explain the motivation and impact of parameters such as $\alpha_t$, the choice of the model $\omega_x^{M,\bar{\delta}}$, and the technique of estimating sequence on the algorithm's performance and convergence.
2) When comparing computational time, is the proposed algorithm better than existing methods?
3) The figures in the paper are too small, making it difficult for readers to interpret the results.

**Details Of Ethics Concerns:**

None.

---

> ### Author Response · Authors · 2023-11-14
>
> Thank you for your helpful feedback and thoughtful inquiries!
>
> >**Question 1:** *“The paper lacks an intuitive explanation of the proposed algorithm. It is suggested to explain the motivation and impact of parameters such as αt, the choice of the model $\omega_{x, M}^{\bar \delta}$ and the technique of estimating sequence on the algorithm's performance and convergence.”*
>
> **Response:** Kindly examine the common commentary titled "On intuition behind the algorithm."
>
> >**Question 2:** *“When comparing computational time, is the proposed algorithm better than existing methods?”*
>
> **Response:** In computational time comparisons, the efficiency is often contingent on the implementation rather than solely on the algorithms themselves. The relative speed of the proposed algorithm compared to existing methods can vary depending on the specific implementation and the characteristics of the minimization problem. To ensure a fair evaluation, we have presented our results in terms of iteration complexity.
>
> When comparing our implementations of SGD, ExtraNewton (EN), and Accelerated Stochastic Cubic Newton (ASCN), all implemented using PyTorch 2.0, in terms of computation time, we observe a slight advantage of ASCN over EN. Notably, both ASCN and EN outperform SGD in scenarios with small Hessian batch sizes ranging from 150 to 1000. We are open to incorporating these additional experiments into the revised version of the paper.
>
> >**Question 3:** *“The figures in the paper are too small, making it difficult for readers to interpret the results.”*
>
> **Response:** Thank you for your feedback regarding the figures in our paper. We acknowledge your concern about their size impacting the ease of interpretation for readers. In the revised version, we will try to address this by resizing the figures for better visibility and comprehension. However, it's essential to note that we are constrained to 9 pages, and any adjustments to figure sizes may necessitate the removal of other elements from the work to maintain brevity. Alternatively, we are open to adding figures from Section 7 to the Appendix, presenting them with increased size as shown in Figures 3 and 4. Your further suggestions on this ma tter are appreciated.

---

> ### Author Response · Authors · 2023-11-22
>
> Dear Reviewer 2pJU,
>
> We have submitted a revised version of our work. In this iteration, we have made several enhancements to address your feedback.
>
> Firstly, we have included running time experiments in Appendix C (Figures 7 and 8). Our findings indicate that. in a setup with a gradient batch size of 10,000, and Hessian batch sizes of 150 and 450, Accelerated Stochastic Cubic Newton outperforms both Extra-Newton and SGD.
>
> Additionally, we have increased the font size for Figures 1 and 2. To provide further clarity and accessibility, we have added a larger version of these graphics to Appendix C (Figures 5 and 6).
>
> Furthermore, we have included a discussion on the intuition behind our method to Appendix B.
>
> We appreciate your valuable insights. Thank you for your time and consideration.

---

### Official Review · Reviewer_S1GY · 2023-10-31

**Soundness:** 2 fair
**Presentation:** 2 fair
**Contribution:** 2 fair
**Rating:** 1
**Confidence:** 5

**Summary:**

The authors present a new accelerated stochastic second-order method that is robust to both gradient and Hessian inexactness, typical in machine learning. It looks to achieve the lower bounds.

**Strengths:**

None

**Weaknesses:**

The algorithm of step 4 also includes another optimization problem.

**Questions:**

How does the algorithm work in step 4 of the proposed algorithm?  Does it work in practice?

---

> ### Author Response · Authors · 2023-11-13
>
> >**Weaknesses :** *"The algorithm of step 4 also includes another optimization problem."*
>
> >**Questions:** *"How does the algorithm work in step 4 of the proposed algorithm? Does it work in practice?"*
>
>
> **The subproblem in line 4 of Algorithm 1 admits a closed form solution**,
> so it cannot be considered as a separate optimization problem. We apologize for not presenting it explicitly in the paper. Allow us to rectify that here.
>
> First, note that the problem in line 4 is convex. Let us write optimality condition
> $$0 = \nabla \psi_t(y_t) = (\lambda_{t}  + \kappa_2^{t} + \kappa_3^{t} ||y_{t} - x_0||)(y_{t} - x_0) + \sum \limits_{j=0}^{t-1} \frac{\alpha_j}{A_j} g(x_{j+1}).$$
> Thus,
> $$(\lambda_{t}  + \kappa_2^{t} + \kappa_3^{t} ||y_{t} - x_0||)||y_{t} - x_0||  =  || \sum \limits_{j=0}^{t-1}\frac{\alpha_j}{A_j} g(x_{j+1})||.$$
> Let us denote $r_{t} = ||y_{t} - x_0||$ and $S_t = || \sum \limits_{j=0}^{t-1}\frac{\alpha_j}{A_j} g(x_{j+1})||$. Then, we get
> $$\kappa_3^{t} r_{t}^2 + (\lambda_{t}  + \kappa_2^{t})r_{t} - S_t = 0.$$
> Next, we get the solution of quadratic equation
> $$r_{t} = \frac{\sqrt{(\lambda_{t}  + \kappa_2^{t})^2 + 4 \kappa_3^{t}S_t } - (\lambda_{t} + \kappa_2^{t} )}{2\kappa_3^{t}}.$$
> Finally, we get explicit solution
>
> $$y_{t} = x_0 - \frac{\sum \limits_{j=0}^{t-1}\frac{\alpha_j}{A_j} g(x_{j+1}) }{\lambda_t + \bar{\kappa}_2^t + \bar{\kappa}_3^t r_t}.$$
>
> We will add this derivation to the revised version of the paper.
>
>
> Furthermore, we would like to point out that *solving minimization subproblems are in the heart of  algorithm design*. For example, even the iconic gradient descent scheme stems directly from minimizing quadratically regularized 1st order Taylor approximation sub-problem:
> $$x_{k+1} = \arg \min_x  (f(x_k) + \langle \nabla f(x_k), x- x_k\rangle + \frac{1}{2\gamma}||x - x_k||^2). $$
>  Gradient descent allows for closed form solution as a subproblem in line 4 in Algorithm 1 . Subproblems like line 4 in Algorithm 1 naturally arise from acceleration techniques. See line 1 of (4.8) from [1], lines 6-7 of Algorithm 1 from [2].  Furthermore, the absence of a closed-form solution for a subproblem should not be viewed as a deterrent when considering a specific optimization method, such as Mirror-Prox [3], Frank-Wolfe methods [4, 5], or Trust-Region methods [5].
>
> **Practical performance**
>
>
> As we have established, the practical performance of subproblem 4 is not an issue. We use the explicit solution in our implementation of the algorithm. We provide numerical evidence for the overall algorithm in section 7. The proposed method works well in practice, one can see the behavior on Figures 1, 2.
>
>
> *Having outlined the specifics of the subproblem in line 4 of Algorithm 1 along with its explicit solution, we wonder whether this might be the reason for the pronounced strong rejection.*
>
> [1] Nesterov, Yu. "Accelerating the cubic regularization of Newton’s method on convex problems." Mathematical Programming 112.1 (2008): 159-181.
>
> [2] Allen-Zhu, Zeyuan, and Lorenzo Orecchia. "Linear coupling: An ultimate unification of gradient and mirror descent." arXiv preprint arXiv:1407.1537 (2014).
>
> [3] Nemirovskij, Arkadij Semenovič, and David Borisovich Yudin. "Problem complexity and method efficiency in optimization." (1983).
>
> [4] Jaggi, Martin. "Revisiting Frank-Wolfe: Projection-free sparse convex optimization." International conference on machine learning. PMLR, 2013.
>
> [5] Nocedal, Jorge, and Stephen J. Wright, eds. Numerical optimization. New York, NY: Springer New York, 1999.

---

> > ### Author Response · Authors · 2023-11-21
> >
> > Thank you for your time on the initial review.
> > Could you please confirm whether our responses clarified your concerns, or if there is more to consider? Your response and feedback  are valuable to us.

---

> > > ### Author Response · Authors · 2023-11-21
> > >
> > > Dear Reviewer S1GY,
> > >
> > > We have submitted a revised version of the paper. Changes have been highlighted in blue. We have included the derivation of subproblem's solution to Appendix H.
> > >
> > > Thank you once again for the feedback.

---

### Official Review · Reviewer_KkLj · 2023-11-02

**Soundness:** 3 good
**Presentation:** 3 good
**Contribution:** 3 good
**Rating:** 8
**Confidence:** 3

**Summary:**

This paper present a stochastic second-order method  based on nesterov acceleration and cubic newton, which is proven to be robust to gradient and Hessian inexactness. The faster convergence rate of this algorithm is established for stochastic Hessian and inexact Hessian compared with previous work. The lower bound for the stochastic/inexact second-order methods for convex function with smooth gradient and Hessian is also established, verifying the tightness of their convergence upper bound. The inprecision produced by solving the cubic subproblem is also taken account of in the analysis. The method is also extended to higher-order minimization with stochastic/inexactness, and a restrated accelerated stochastic tensor method is also proposed for strongly-convex function.

**Strengths:**

1. The author of this article considers several interesting questions that naturally arise for the inexactness in practice, and are therefore valuable, especially the proposed algorithm and its tight convergence rate, and the lower bounds for inexact second-order methods.
2. The article follows a natural logic in exploring the questions,  progressing in a layered manner.
3. The proof techniques the authors used seem solid and sound.

**Weaknesses:**

1. While I grasp the overarching concept of the algorithm and the principal steps in the proof, the exposition doesn't offer much in the way of intuitive understanding. Could the authors elucidate the rationale behind the algorithm's design and the parameter choices? Enhancing the exposition with additional intuitive insights into the algorithm would be highly beneficial.
2. The assumptions (2.2 and 2.3) of stochacity and inexactness differ but seem highly related, as they lead to two quite similar convergence rates and proof for your algorithm. In this paper the way of discussing the stochasticity and inexactness settings seems a bit nesting.  Maybe it could be better if their highlevel relations  are set forth in a proper manner.
3. Clarification: O(1/T^2) ->\Omega(1/T^2) in P2 line 2; bounded variance stochastic Hessian ->stochastic Hessian with bounded variance; formatting issues such as sequence numbers in the algorithm list.

**Questions:**

1. In section 3, the subproblem you defined is $\omega_{x}^{M,\bar{\delta}}=f\left(x\right)+\left\langle g\left(x\right),y-x\right\rangle +\frac{1}{2}\left\langle y-x,H\left(x\right)\left(y-x\right)\right\rangle +\frac{\bar{\delta}}{2}\left\Vert x-y\right\Vert ^{2}+\frac{M}{6}\left\Vert x-y\right\Vert ^{3}$. Compared to the original cubic regularized subproblem, in addition to the modification of the inexactness and stochasticity, your formulation has an additional quadratic term $\frac{\bar{\delta}}{2}\left\Vert x-y\right\Vert ^{2}$. Are there any reason or intuition for this term?

2. Could you bring more insights for the aggregation of stochastic linear models above algorithm 1?

3. The gloabal convergent (accelerated, cubic newton type) second-order methods, while they have accelerated global convergence, they usually can be proven to have superlinear local convergence rate. Is the parallel local characteristic worth to be mentioned and investigated in your stochastic/inexact setting?

4. The open question mentioned, "what's the optimal trade-off between inexactness in gradients and the Hessian?", where in the article is intuitively investigated?

5. Regarding Algorithm 3, you mentioned the 'Restarted Accelerated Stochastic Tensor Method'. Could you further elaborate on the specific mechanism and necessity of this 'restarting'? Under what circumstances should a restart be performed?

---

> ### Author Response · Authors · 2023-11-14
>
> Thank you for your insightful feedback and valuable questions!
>
> >**Weakness 1:** *“While I grasp the overarching concept of the algorithm and the principal steps in the proof, the exposition doesn't offer much in the way of intuitive understanding. Could the authors elucidate the rationale behind the algorithm's design and the parameter choices? Enhancing the exposition with additional intuitive insights into the algorithm would be highly beneficial.”*
>
> **Response:**  Please review the common commentary “On the intuition behind the algorithm”.
>
> >**Weakness 2:** *“The assumptions (2.2 and 2.3) of stochacity and inexactness differ but seem highly related, as they lead to two quite similar convergence rates and proof for your algorithm. In this paper the way of discussing the stochasticity and inexactness settings seems a bit nesting. Maybe it could be better if their highlevel relations are set forth in a proper manner.”*
>
> **Response:**
> We appreciate your observation regarding the nesting of Assumptions (2.2 and 2.3) and their apparent similarities in leading to analogous convergence rates and proof for our algorithms. While we recognize their close relationship, it's crucial to emphasize a fundamental distinction between the two.  Assumption 2.2 imposes a bound on the expected norm of the difference between the Hessian and the stochastic Hessian, whereas Assumption 2.3 pertains to the difference along the direction $y - x$. Consequently, 2.3 is necessary only for iterations of the Algorithm, and the value of $\delta_2^{x, y}$ can be significantly smaller than the norm of the difference between the Hessian and its approximation. This delineation underscores why we present both assumptions and provide separate theoretical results for each. If you believe that presenting both assumptions is unnecessary, we welcome further discussion on this matter.
>
> >**Weakness 3:** *“Clarification: O(1/T^2) ->\Omega(1/T^2) in P2 line 2; bounded variance stochastic Hessian ->stochastic Hessian with bounded variance; formatting issues such as sequence numbers in the algorithm list.”*
>
> **Response:**  Thank you for bringing that to our attention; we will make the necessary changes to the paper in the revised version.
>
> >**Question 1:** *“In section 3, the subproblem you defined is $ω_{x, M}^{\bar δ}(y)=f(x)+⟨g(x),y−x⟩+\frac{1}{2}⟨y−x,H(x)(y−x)⟩+\frac{\bar δ}{2}‖x−y‖^2+\frac{M}{6}‖x−y‖^3$. Compared to the original cubic regularized subproblem, in addition to the modification of the inexactness and stochasticity, your formulation has an additional quadratic term $\frac{\bar \delta}{2}‖x−y‖^2$. Are there any reason or intuition for this term?.”*
>
> **Response:** Please have a look at the part A of  common commentary “On the intuition behind the algorithm”.
>
> >**Question 2:** *“Could you bring more insights for the aggregation of stochastic linear models above algorithm 1?”*
>
> **Response:** Please have a look at the part B of  common commentary “On the intuition behind the algorithm”.
>
> >**Question 3:** *“The gloabal convergent (accelerated, cubic newton type) second-order methods, while they have accelerated global convergence, they usually can be proven to have superlinear local convergence rate. Is the parallel local characteristic worth to be mentioned and investigated in your stochastic/inexact setting?”*
>
> **Response:** Thank you for your insightful commentary.
>
> While local convergence seems to be an interesting theoretical direction, it lies beyond the scope of this paper, which primarily concentrates on global convergence without assumptions on initial point. Local convergence normally requires some very restrictive assumptions on starting point and batch size (gradient accuracy) which are hard to satisfy in practice.
>
> Illustratively, consider the local superlinear convergence from [1], which requires $f(x_0) - f(x^*) \leq \frac{\mu^3}{L_2^2}$. In most practical scenarios desired accuracy $\varepsilon \lesssim  \frac{\mu^3}{L_2^2}$.
>
> In summary, while the local convergence of stochastic/inexact accelerated second-order/high-order methods remains an open question, we believe that exploring this aspect may not make the proposed scheme much more practical. Consequently, we maintain our focus on global performance within the defined scope of this work.
>
> >**Question 4:** *“The open question mentioned, "what's the optimal trade-off between inexactness in gradients and the Hessian?", where in the article is intuitively investigated? ”*
>
> **Response:** We establish the lower bound $\Omega(\frac{\sigma_1 R}{\sqrt{T}} + \frac{\sigma_2 R^2}{T^2} + \frac{L_2 R^{3}}{T^{7/2}})$, which shows that the best possible tradeoff between inexactness is $\Omega(\frac{\sigma_1 R}{\sqrt{T}} + \frac{\sigma_2 R^2}{T^2})$, which is achieved by proposed algorithm.
>
> [1] Kovalev, Dmitry, Konstantin Mishchenko, and Peter Richtárik. "Stochastic Newton and cubic Newton methods with simple local linear-quadratic rates." arXiv preprint arXiv:1912.01597 (2019).

---

> > ### Author Response · Authors · 2023-11-14
> >
> > >**Question 5:** *“Regarding Algorithm 3, you mentioned the 'Restarted Accelerated Stochastic Tensor Method'. Could you further elaborate on the specific mechanism and necessity of this 'restarting'? Under what circumstances should a restart be performed?”*
> >
> > **Response:** Restarts serve as a technique to adapt a method designed for convex problems to those that are strongly convex. Typically, restarts ensure the preservation of optimality, meaning that an optimal method for convex functions can be extended to become an optimal method for strongly convex ones. The restart occurs after a predetermined number of iterations, as outlined in Algorithm 3 and (15).
> >
> > Our presentation of the restarted method is motivated not only by its applicability to strongly-convex problems but also by a compelling observation. Specifically, the overall number of stochastic Hessian computations, $O(\frac{\sigma_2}{\mu^{1/3}} )\log \frac{1}{\varepsilon}$ scales linearly with the desired accuracy $\varepsilon$, as elucidated in the paragraph following Theorem 6.2. This noteworthy insight enables the use of constant batch sizes for stochastic Hessian, independent of $\varepsilon$. This stands in contrast to first-order derivatives, where the batch size is dependent on $\varepsilon$.

---

> > > ### Author Response · Authors · 2023-11-21
> > >
> > > Dear Reviewer KkLj,
> > >
> > > We have uploaded an updated version of the paper, where changes are marked in blue. We believe that we have addressed all the questions pointed out by you. Additionally, we have included a section explaining the intuition behind the method in Appendix B.
> > >
> > > Thank you once again for your helpful feedback.

---

### Author Response · Authors · 2023-11-14
**On the intuition behind the algorithm**

Dear Reviewers,

In the following, we endeavor to provide an interpretation of the proposed methodology.

A. **Model**

For the second-order case the model $\omega_{x, M}^{\bar{\delta}}(y)$ comprises three key components: an inexact Taylor approximation $\phi_x(y)$; cubic regularization $\frac{M}{6}||x-y||^3$ and additional quadratic regularization $\frac{\bar{\delta}}{2}||x-y||^2$.
The combination of Taylor polynomial and cubic regularization is the standard model for exact second-order methods, as they create a model that is both convex and upper bounds the objective [2] (see [3] for high-order optimization). However, inserting inexactness to the Taylor approximation leads to the necessity of additional regularization [4].

The first reason to add quadratic regularization is to ensure that the Hessian of the function is majorized by the Hessian of the model:

$0 \preceq \nabla^2 f(y) \preceq \nabla^2 \phi_x(y) + \delta_2 I + L_2||y - x|| \preceq \nabla^2 \phi_x(y) + \bar{\delta}_2 I + M||y - x|| = \nabla^2 \omega_x^{\bar{\delta}}(y)$

Moreover, this regularization is essential for handling stochastic gradients correctly. Note, that we add quadratic regularizer with the constant $\bar{\delta}_t = 2\delta_2 + \frac{\sigma_1}{R}(t+1)^{3/2}$. Here, $\delta_2$ accounts for a Hessian majorization, while $\frac{\sigma_1}{R}(t+1)^{3/2}$ is crucial for achieving optimal convergence in gradient inexactness. From our perspective, this regularization can be viewed as a damping for the size of stochastic Cubic Newton step, as stochastic gradients may lead to undesirable directions.

For further clarification, please refer to Lemma D.3 in the Appendix. This lemma serves as a bound on the progress of the step. Take a look at the right-hand side of equation (26). Without proper quadratic regularization, we won't capture the correct term related to Hessian inexactness $\delta_2$. Consequently, the desired convergence term, $\frac{\delta_2 R^2}{T^2}$, cannot be achieved. Moving on to the left-hand side of (26), we encounter the term $\frac{2}{\bar{\delta}}||g(x) - \nabla f(x)||$. Here, choosing the appropriate $\bar{\delta}$ is crucial to compensate for stochastic gradient errors and achieve optimal convergence in the gradient inexactness term.

B. **Estimating sequences**

Estimating sequences are a standard optimization technique to achieve acceleration (Section 2.2.1 [1]). As far as our knowledge extends, the application of estimating sequences to second-order methods was first introduced in [2]. The concept involves adapting acceleration techniques traditionally applied to first-order methods to the realm of second-order methods. In this work, we make slight modifications to the estimating sequences derived from [2] to preserve the customary relationships inherent in accelerated methods:

$\frac{f(x_t)}{A_{t-1}} - err_{low} \leq \psi_t^*$

$\psi_t^* \leq \frac{f(x^*)}{A_{t-1}} + c_2 ||x^* - x_0||^2 + c_3 ||x^* - x_0||^3 + err_{up}.$

$A_{t}$ and $\alpha_t$ are scaling factors, common for acceleration [1, 2].

For simplicity, let $err_{low} = 0, ~ err_{up} = 0, ~ c_2 = 0$. That is the case for exact derivatives and subproblem solutions. Then, one can get the convergence, with a specific choice of scaling factors.
$f(x_t) - f(x^*) \leq A_{t-1} c_3 ||x^* - x_0||^3.$

In our case, errors and $c_2$ are non zero and stay for gradient, Hessian and subproblem inexactness. By applying estimating sequence technique we get the rate (9-10).

C. **The choice of parameters**

The cubic regularization parameter $M \geq 2 L_2$ represents the standard choice for second-order methods [2].

The quadratic regularization parameter  $\bar{\delta}_t = 2\delta_2 + \frac{\sigma_1 + \tau}{R}(t+1)^{3/2}$ consists of  $2 \delta_2$ for compensating Hessian errors, and $\frac{\sigma_1 + \tau}{R}(t+1)^{3/2}$ for compensating stochastic gradient and subproblem solution errors.

The regularization parameters $\bar{\kappa}_2^t, \bar{\kappa}_3^t, \lambda_t$ are utilized for the second step of the method. The $\bar{\kappa}$’s are chosen in (37, 39) to uphold the inequality (31) for acceleration. $\lambda_t = \frac{\sigma}{R}(t+1)^{5/2} $ serves to compensate stochastic gradient errors in the estimation functions  $\psi_t$. The specific choice for $\bar{\delta}$ and $\lambda$ is made in the proof of Theorem E5 to achieve optimal convergence rate.


*References:*

[1] Nesterov, Yurii. Lectures on convex optimization. Vol. 137. Berlin: Springer, 2018.

[2] Nesterov, Yurii, and Boris T. Polyak. "Cubic regularization of Newton method and its global performance." Mathematical Programming 108.1 (2006): 177-205.

[3] Nesterov, Yurii. "Implementable tensor methods in unconstrained convex optimization." Mathematical Programming 186 (2021): 157-183.

[4] Agafonov, Artem, et al. "Inexact tensor methods and their application to stochastic convex optimization." arXiv preprint arXiv:2012.15636 (2020).

---

### Meta-Review · Area_Chair_sdd3 · 2023-12-21

**Metareview:**

The paper considers the problem of convex minimization and is concerned with second(or higher order) minimization when the gradient and Hessian information is inexact/stochastic. In particular they consider the case of stochastic (bounded variance) gradient oracle and a stochastic Hessian where in expectation the spectral norm of the difference between the true Hessian and the staochastic Hessian is small. There are two main results in the paper

1. An upper bound on the convergence rate where in the term that is dependent on the inexactness level of the Hessian is improved (in terms of the dependence on the number of steps)
2. A lower bound which establishes that the paper's results are optimal in the settings it considers.

Overall both the paper's contribution are clear and concise and interesting to the optimization community. The paper writing although can definitely use multiple revisions esepecially to bring out salients aspects of the their analysis. There are a few demerits of the paper for eg. there is limited algorithmic contribution in the sense that the paper uses the well known Cubic Regularization Method and the main contribution is that it is stable under these kinds of errors. The method is also very applicable in any anything beyond small scale settings for machine learning.

Overall I find the paper borderline but reviewers appreciated the contributions of the paper which are clear in an optimization context and I am therefore leaning towards an accept.

**Justification For Why Not Higher Score:**

The contributions are clearly limited in terms of the algorithm and also the proposed algorithm is not practical for beyond small scale settings.

**Justification For Why Not Lower Score:**

I believe the paper is a borderline paper however the reviewers have appreciated the clear contributions both in terms ofupper and lower bound and the closing of the gap between for this problem setting.

---

### Decision · Program_Chairs · 2024-01-16

Accept (poster)